# Gated Domain Units for Multi-source Domain Generalization

**Simon Föll**[1*]                                                          *sfoell@ethz.ch*

**Alina Dubatovka**[2*]                                         *alina.dubatovka@inf.ethz.ch*

**Eugen Ernst**[3†]                                                      *euernst@kit.edu*

**Siu Lun Chau**[5†]                                            *siu-lun.chau@cispa.de*

**Martin Maritsch**[1]                                           *mmaritsch@ethz.ch*

**Patrik Okanovic**[2]                                             *pokanovic@ethz.ch*

**Gudrun Thäter**[3]                                         *gudrun.thaeter@kit.edu*

**Joachim M. Buhmann**[2]                                     *jbuhmann@inf.ethz.ch*

**Felix Wortmann**[4]                                        *felix.wortmann@unisg.ch*

**Krikamol Muandet**[5]                                          *muandet@cispa.de*

[1] *Department of Management, Technology, and Economics, ETH Zurich, Switzerland*
[2] *Department of Computer Science, ETH Zurich, Switzerland*
[3] *Department of Mathematics, Karlsruhe Institute for Technology, Germany*
[4] *Institute for Technology Management, University of St. Gallen, Switzerland*
[5] *CISPA–Helmholtz Center for Information Security, Germany*

**Reviewed on OpenReview:** `https://openreview.net/forum?id=V7BvYJyTmM`

## Abstract

The phenomenon of distribution shift (DS) occurs when a dataset at test time differs from the dataset at training time, which can significantly impair the performance of a machine learning model in practical settings due to a lack of knowledge about the data's distribution at test time. To address this problem, we postulate that real-world distributions are composed of latent *Invariant Elementary Distributions* (I.E.D) across different domains. This assumption implies an invariant structure in the solution space that enables knowledge transfer to unseen domains. To exploit this property for domain generalization, we introduce a modular neural network layer consisting of Gated Domain Units (GDUs) that learn a representation for each latent elementary distribution. During inference, a weighted ensemble of learning machines can be created by comparing new observations with the representations of each elementary distribution. Our flexible framework also accommodates scenarios where explicit domain information is not present. Extensive experiments on image, text, and graph data show consistent performance improvement on out-of-training target domains. These findings support the practicality of the I.E.D assumption and the effectiveness of GDUs for domain generalisation.

---

[*]SF and AD contributed equally to this paper.
[†]EE and SLC contributed equally to this paper.

# 1 Introduction

Machine learning relies on the fundamental assumption that training and test data are independently and identically distributed (I.I.D.), which ensures the learning machine attains the lowest achievable risk as the sample size grows under the empirical risk minimization (ERM) framework (Vapnik, 1998; Schölkopf, 2019). However, numerous research and real-world applications (Zhao et al., 2018; 2020; Ren et al., 2019; Taori et al., 2020) have provided staggering evidence against this assumption; as shown in D'Amour et al. (2020). In practice, the I.I.D. assumption is often violated due to distribution shift (DS), which occurs where a model is applied to a dataset that differs from its training data (Sugiyama & Kawanabe, 2012), resulting in significantly impaired performance.

To tackle DS, recent work advocates for domain generalization (DG): Given data from multiple domains, e.g., Continental Europe hospitals, how to train a model that can generalize well to unseen domains, e.g., US hospital? (Blanchard et al., 2011; Muandet et al., 2013; Zhou et al., 2021a). The ability to generalize to entirely new domains is crucial for the robust and safe deployment of machine learning models, particularly when unforeseeable domains arise after deployment. Nevertheless, the question of how to identify the appropriate *invariance* that enables such generalization remains an open and unresolved issue, as noted by Gulrajani & Lopez-Paz (2020).

In the following, we introduce two examples from Koh et al. (2021a) to challenge common assumptions of domain definitions from the well-established benchmark for DG. In *camelyon17* (Bándi et al., 2019), the task is to train a model to classify a tumor (i.e., benign or malign) based on images of tissue slices from a few hospitals and test the model on an unseen hospital. Here, each hospital is considered a domain. However, when studying tissue slices under a microscope, the source of variation arises from differences in patient population, slide staining, and image acquisition, and thus not necessarily from the hospital itself Veta et al. (2016); Komura & Ishikawa (2018); Tellez et al. (2019). Further, in *OGB-MolPCBA* (Hu et al., 2021), the task is to predict the biochemical properties of small molecules from a set of possible molecules and generalize to a set of molecules structurally different from those seen in the training set. Each scaffold (i.e., subset of structurally similar molecules) is considered a domain, thus yielding over 120,000 domains in *OGB-MolPCBA*. Though the definition of domains is bio-chemically motivated, this assumption causes computation overhead, making it challenging to find an invariance across these scaffolds.

Prior work has made a similar motivation to advocate for learning so-called latent domains in the combined source dataset through an unsupervised manner Deecke et al. (2022); Matsuura & Harada (2020); Chen et al. (2022a). The motivation is to become independent of domain labels, when applying DG that learn an invariant structure across domains during training (see Section 2 for a discussion). However, this line of work does not draw a connection between the latent source domains and test domains. In contrast, our paper follows the current trend in DG research that establishes connections between test-time and source domains. Unlike previous work that assumes test-time domains are convex combinations of source domains, we propose that both domains are linear combinations of some latent invariant units, which can be learned during training. This assumption, formally introduced in Definition 1, induces an invariant structure in the hypothesis space, which are practically appealing for domain generalization (cf. Proposition 3.1). To turn theory into practice, we develop a neural network layer consisting of gated domain units (GDUs) that learn a geometric representation of each elementary domains in the form of kernel mean embedding (KME) (Berlinet & Thomas-Agnan, 2004; Smola et al., 2007; Muandet et al., 2017). During inference, a weighted ensemble of learning machines can be created by comparing new observations with the elementary domains embeddings. We depicted our idea in Figure 1.

We summarise our contributions as follows:

1. We propose the Invariant Elementary Distribution (I.E.D) assumption, postulating both test-time and source domains consist of latent elementary distributions that can be learned during training, and show the assumption's practical appeal for domain generalization.

2. We develop a modular neural network layer consisting of Gated Domain Units, each captures and learns an elementary distribution via kernel mean embeddings. These representations can thus be used to adapt the ensemble weights to an unseen domain at test time.

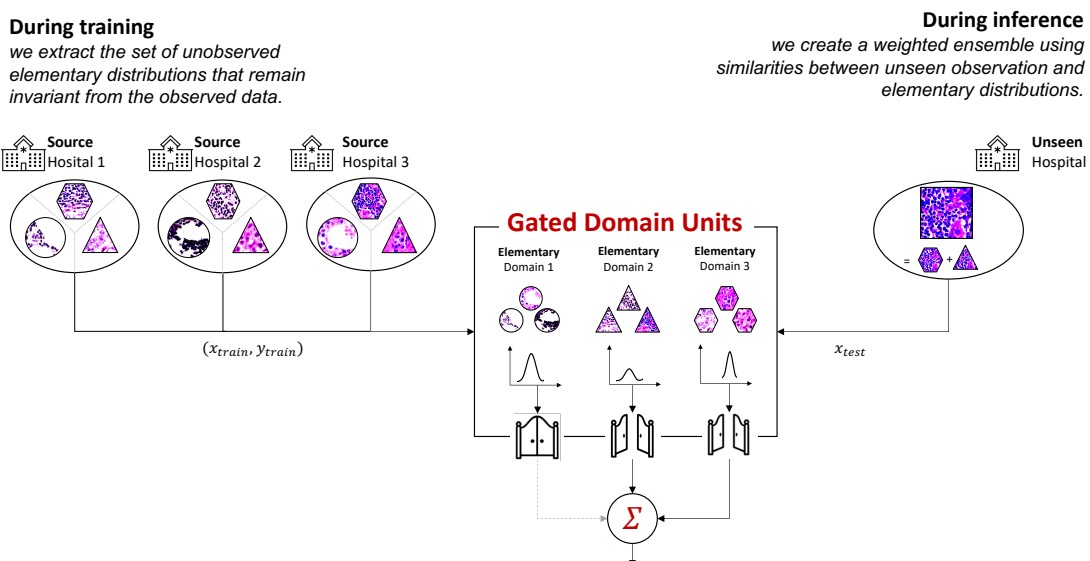

Figure 1: An illustration of invariant elementary distributions in *camelyton17* dataset and the concept of the Gated Domain Units (GDUs). While data sources are often considered as domains, each hospital contains heterogene subsets that are shared across sources. We can depict examples of these subsets (latent domains) in three different shapes (rectangle, octahedron, and circle) for each domain following the method of Matsuura & Harada (2020). In contrast to existing work, the invariant elementary distribution (I.E.D.) assumption advocates that these latent domains remain invariant across the hospitals. Therefore, we propose to learn representations of these elementary domains akin to mixture models, and utilise our Gated Domain Units to build a weighted ensemble during inference time.

3. We verify the I.E.D assumption by extensive experiments using a publicly available benchmarking WILDS. Specifically, we validate our method on image, text, and graph datasets, showing consistent improvement on out-of-training target domains.

4. We provide an effective TensorFlow and PyTorch implementation applicable to different feature extractors such as ResNet50, DistillBERT, and GIN virtual and thus a broad audience of researchers and practitioners to enable a fast adaption of our method.

The remainder of this paper is organized as follows: Section 2 outlines related work, and lays out research gaps that we aim to address. Our theoretical framework is presented in Section 3, followed by our modular Gated Domain layer implementation shown in Section 4. Our experimental evaluations are then presented in Section 5. Finally, we discuss the potential limitations and future work in Section 6.

## 2 Previous Work and Motivation

DG is among the hardest problems in machine learning (Blanchard et al., 2011; Muandet et al., 2013; Arjovsky et al., 2019). In contrast, domain adaptation (DA), which predates DG, deals with a slightly simpler scenario in which some data from the test distribution are available (Ganin et al., 2015). Hence, based on the available data, the task is to develop learning machines that transfer knowledge learned in a source domain specifically to the target domain. Approaches pursued in DA can be grouped primarily into (1) discrepancy-based DA (Sun et al., 2016; Peng & Saenko, 2018; Ben-David et al., 2010; Fang et al., 2020; Tzeng et al., 2014; Long et al., 2015) (2) adversary-based DA (Tzeng et al., 2017; Liu & Tuzel, 2016; Ganin et al., 2015; Long et al., 2018), and (3) reconstruction-based DA (Bousmalis et al., 2016; Hoffman et al., 2018b; Kim et al., 2017; Yi et al., 2017; Zhu et al., 2017; Ghifary et al., 2014). In DA, learning the domain-invariant components requires access to unlabeled data from the target domain. Unlike problems in DA, where the unlabeled data from

the test domains is accessible to find the right invariant structures (Ben-David et al., 2010), the lack thereof in DG calls for a *postulation* of invariant structure that will enable generalization (Gulrajani & Lopez-Paz, 2020).

To enable generalization to unseen domains without any access to data from them, researchers have made significant progress in the past decade and developed a broad spectrum of methodologies (Zhou et al., 2021a;b; Li et al., 2019; Blanchard et al., 2011). For thorough review see, e.g., Zhou et al. (2021a); Wang et al. (2021). Existing works can be categorized into methods based on domain-invariant representation learning (Muandet et al., 2013; Li et al., 2018b;c), meta-learning (Li et al., 2018a; Balaji et al., 2018), data augmentation (Zhou et al., 2020), distributionally robust optimization Dai et al. (2023); Sagawa et al. (2020), to name a few. Another recent stream of research from a causal perspective includes invariant risk minimization (Arjovsky et al., 2019), invariant causal prediction (Peters et al., 2016), and causal representation learning (Schölkopf et al., 2021; Chen et al., 2022b). The overall motivation here is to learn the representation that is robust to domain-specific spurious correlations. In other words, it is postulated that "causal" features are the right kind of invariance that will enable generalization. On the other hand, latent domain discovery is another popular approach in DG. Finding pseudo-domains through an unsupervised manner allows one to learn invariant feature extractors across such latent domains (Hoffman et al., 2012; Li et al., 2020; Gong et al., 2013; Matsuura & Harada, 2020). This line of work is practically appealing when collecting explicit domain labels are difficult in practice, as discussed in Appendix B.2.2.

Finally, ensemble learning has been shown to be effective in DG (Wang et al., 2021). A particular branch of ensemble learning is domain-specific models, which aim to learn a source domain-specific model, i.e., models specializing in each source domain (Piratla et al., 2020; Monteiro et al., 2021). When determining the weighting for an ensemble, it has been shown that making the weights adaptive during inference is advantageous instead of just averaging the prediction. This adaptive weighting adjusts the ensemble weights during inference based on how similar each source domain is to the test domain Zhang et al. (2023); Gao et al. (2008); Yao & Doretto (2010). However, these domain-specific models require domain labels to train the domain classifiers and additional models, making it challenging to integrate the approach into a holistic and efficient framework. Further, and most importantly, these approaches have not been combined with the idea of latent elementary domains.

**Motivation.** Our work focuses on addressing the following two challenges to generalize the idea of latent domains. First, previous work has not considered the invariance of latent representations between source and test domains, which limits their effectiveness for test-time adjustments and adaptation. To address this, we introduce the I.E.D assumption. Second, creating effective architectures for these approaches can be difficult, especially ones that can seamlessly integrate with existing deep learning frameworks. To overcome this challenge, we introduce our Gated Domain Units, which leverage the I.E.D assumption. Practitioners require methods that can handle feature extractors for various data types, such asResNet-50 (He et al., 2016), DenseNet-121 Huang et al. (2017), Graph Isomorphism Networks with virtual nodes (GIN-virtual; Xu et al. (2019) and Gilmer et al. (2017)), DistillBERT Sanh et al. (2020), with minimal integration effort. In our work, we propose the "invariant elementary distributions" assumption to tackle the first challenge, and we propose an effective layer consisting of Gated Domain Units to address the second challenge. More details on this can be found in the next two sections.

## 3 Domain Generalization with Invariant Elementary Distributions

We leverage the above observation for domain generalization by considering the *mixture component shift* - the most common form of DS - which states that the data is made up of different sources, each with its own characteristics, and their proportions vary between domains (Quinonero-Candela et al., 2022, pp. 19).

### 3.1 Invariant Elementary Distributions

Let $\mathcal{X}$ and $\mathcal{Y}$ be the input and output space, with a joint distribution $\mathbb{P}$. Let $X$ and $Y$ be random variables taking values in $\mathcal{X}$ and $\mathcal{Y}$, respectively. We are given a set of $D$ labeled source datasets $\{\mathcal{D}_i^s\}_{i=1}^D$ with $\mathcal{D}_i^s \subseteq \mathcal{X} \times \mathcal{Y}$. Each of the source datasets is assumed to be I.I.D. generated by a joint distribution $\mathbb{P}_i^s$ with

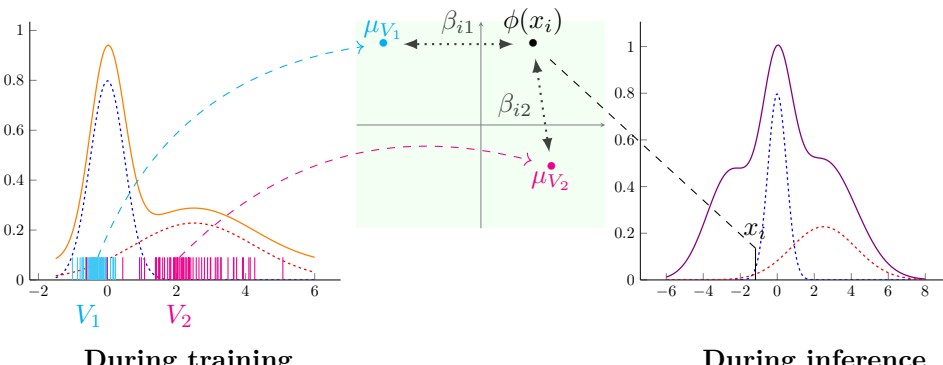

**During training**                                    **During inference**

Figure 2: A visualization of an "invariant elementary distribution (I.E.D.)" assumption for domain generalization (DG): both observed and test-time data distributions (orange and violet) are composed of the same set of *unobserved* elementary distributions (blue and red) that remain invariant across different domains. Hence, the first challenge, during training (left panel), is to extract these elementary distributions from the observed data (orange). The unobserved elementary distributions are represented by the elementary bases $V_1$ and $V_2$ (cyan and magenta). The second challenge, during inference (right panel), is to create a weighted ensemble of learning machines that utilize the similarities between the embedding of the unseen observation $\phi(x_i)$ and the embeddings of these distributions $\mu_{V_1}$ and $\mu_{V_2}$ in the RKHS $\mathcal{H}$ (green rectangle) as weights $\beta_{i1}$ and $\beta_{i2}$.

support on $\mathcal{X} \times \mathcal{Y}$, henceforth denoted *domain*. Note that the conditional distributions of different domains may change. Hence, our work differs from related work in DG that rely on the covariate shift assumption, i.e., the conditional distribution of the training and test data stays the same (David et al., 2010). The set of probability measures with support on $\mathcal{X} \times \mathcal{Y}$ is denoted by $\mathcal{P}$. The multi-source dataset $\mathcal{D}^s$ comprises the merged individual source datasets $\{\mathcal{D}^s_j\}_{j=1}^D$. We aim to minimize the empirical risk, see Section 4.2 for details.

Similar to Mansour et al. (2009; 2012) and Hoffman et al. (2018a), we assume that the distribution of the source dataset can be described as a convex combination $\mathbb{P}^s = \sum_{j=1}^D \alpha_j^s \mathbb{P}_j^s$ where $\alpha^s = (\alpha_1^s, \ldots, \alpha_D^s)$ is an element of the probability simplex, i.e., $\alpha^s \in \Delta^D := \{\alpha \in \mathbb{R}^D \mid \alpha_j \geq 0 \wedge \sum_{j=1}^D \alpha_j = 1\}$. In other words, $\alpha_j^s$ quantifies the contribution of each individual source domain to the joint source dataset. For the test domain, prior work Mansour et al. (2009; 2012) and Hoffman et al. (2018a) further makes the important assumption that it lives in the convex hull of the source domains. We aim to tie on these assumptions and connect them to the idea of latent domains, as will discuss in the following paragraph.

For this connection, we express the distribution of each domain as a convex combination of $K$ *elementary distributions* $\{\mathbb{P}_j^e\}_{j=1}^K \subset \mathcal{P}$, meaning that $\mathbb{P}_j^s = \sum_{j=1}^K \alpha_j^e \mathbb{P}_j^e$ where $\alpha^s \in \Delta^K$. Our main assumption is that these elementary distributions *remain invariant across the domains*. The advantage is that we can find an invariant subspace at a more elementary level as opposed to when we consider the source domains as some sort of basis when generalizing to unseen domains. For a previously unseen distribution, we aim to determine the coefficients $\alpha_j^e$ and quantify the similarity to each elementary domain. Figure 2 illustrates this idea.

**Pareto invariance**   The I.E.D assumption implies the invariant structure in the hypothesis space that can be exploited during training, as shown in the following proposition. The proof is given in Appendix A.1.

Table 1: Notation

| | |
|---|---|
| K | number of elementary distributions |
| M | number of elementary domain bases |
| N | number of basis vectors |
| $\mathbb{P}^s$ | combined multi-source distribution |
| $\mathbb{P}_j^s$ | $j$-th single-source distribution |
| $\mathbb{P}_j^e$ | $j$-th elementary distribution |
| $V_j$ | $j$-th domain basis |
| $v_k^j$ | $k$-th vector in $V_j$ |
| $\alpha_j^s$ | coefficient for $\mathbb{P}_j^s$ |
| $\alpha_j^e$ | coefficient for $\mathbb{P}_j$ |
| $\beta_{ij}$ | coefficient for sample $x_i$ and $\mu_{V_j}$ |

**Definition 1.** *Let $\mathcal{F}$ be a hypothesis space of functions and $(R_1, \ldots, R_K) : \mathcal{F} \to \mathbb{R}_+^K$ a vector of risk functionals for $K \geq 2$. Then, the hypothesis $f \in \mathcal{F}$ is said*

*to be Pareto-optimal w.r.t. $\mathcal{F}$ if there exists no $g \in \mathcal{F}$ such that $R_j(g) \leq R_j(f)$ for all $j \in \{1, \ldots, K\}$ with $R_j(g) < R_j(f)$ for some $j$; see, also, Sener & Koltun (2018, Definition 1).*

In other words, $f \in \mathcal{F}$ is Pareto optimal if there exists no hypothesis $g \in \mathcal{F}$ that optimizes every risk functional $R_j$ better than $f$.

**Proposition 3.1.** *Let $\mathcal{L} : \mathcal{Y} \times \mathcal{Y} \to \mathbb{R}_+$ be a non-negative loss function, $\mathcal{F}$ a hypothesis space of functions $f : \mathcal{X} \to \mathcal{Y}$, and $\mathbb{P}^s(X, Y)$ a data distribution. Suppose the I.E.D assumption holds, i.e., there exist $K$ elementary distributions $\mathbb{P}_1^e, \ldots, \mathbb{P}_K^e$ such that $\mathbb{P}^s = \sum_{j=1}^{K} \alpha_j^e \mathbb{P}_j^e$ for some $\boldsymbol{\alpha}^e \in \Delta^K$. Then, the corresponding Bayes predictor $f^* \in \arg\min_{f \in \mathcal{F}} \mathbb{E}_{(X,Y) \sim \mathbb{P}^s}[\mathcal{L}(Y, f(X))]$ is Pareto-optimal w.r.t. $\mathcal{F}$ and elementary risk functionals $(R_1, \ldots, R_K)$ where $R_j(f) := \mathbb{E}_{(X,Y) \sim \mathbb{P}_j^e}[\mathcal{L}(Y, f(X))]$.*

Proposition 3.1 implies that, under the I.E.D assumption, Bayes predictors must belong to a subspace of $\mathcal{F}$ called the Pareto set $\mathcal{F}_{\text{Pareto}} \subset \mathcal{F}$ which consists of Pareto-optimal models. In other words, the I.E.D assumption allows us to translate the invariance property of data distributions to the hypothesis space. Since Bayes predictors of *all* future test domains must lie within the Pareto set, which is a strict subset of the original hypothesis space, it is sufficient for the purpose of generalization to maintain only solutions within this Pareto set during the training time. This makes the solutions within the Pareto set independent of actual coefficients $\alpha$ of the source distribution and potential target distributions. Unfortunately, neither the elementary distributions nor the weights $\boldsymbol{\alpha}$ are known in practice. Motivated by this theoretical insight, our method presented in Section 4 is designed to uncover them from a multi-source training dataset $\mathcal{D}^s$. Reducing the search space from the entire hypothesis space $\mathcal{F}$ to the Pareto set w.r.t. elementary risk functionals $(R_1, \ldots, R_K)$ suggests a modular structure of our proposed Gated Domain layer that consists of Gated Domain Units and each of these units optimizes one risk functional $R_j$. The proposed decomposition also suggests that the Pareto set is independent on the actual coefficients $\alpha$ which suggests an ensemble-like design of our approach. The final solution is a linear combination of the functions optimizing the elementary risk functionals $(R_1, \ldots, R_K)$ with coefficients $\alpha$ that are derived from the similarity between the target distribution and the elementary distribution.

## 3.2 Kernel Mean Embedding of Elementary Distributions

We employ the kernel mean embedding (KME) (Berlinet & Thomas-Agnan, 2004; Smola et al., 2007; Muandet et al., 2017) to represent the elementary distributions. Let $\mathcal{H}$ be a reproducing kernel Hilbert space (RKHS) of real-valued functions on $\mathcal{X}$ with a reproducing kernel $k : \mathcal{X} \times \mathcal{X} \to \mathbb{R}$ (Schölkopf et al., 2001). The KME of a probability measure $\mathbb{P}$ in the $\mathcal{H}$ is defined by $\phi(\mathbb{P}) = \mu_{\mathbb{P}} := \int_{\mathcal{X}} k(\mathbf{x}, \cdot) \, d\mathbb{P}(\mathbf{x})$. Given I.I.D. samples $\{x_i\}_{i=1}^n$ from $\mathbb{P}$, $\mu_{\mathbb{P}}$ can be approximated by the empirical KME $\hat{\mu}_{\mathbb{P}} = n^{-1} \sum_{i=1}^{n} k(x_i, \cdot)$. KME is popular in practice because no explicit distributional assumption, e.g. normality, is required. As such, it has been applied to various machine learning applications, including hypothesis testing (Gretton et al., 2012), training deep generative models (Li et al., 2017), causal inference (Chau et al., 2021), explainability (Chau et al., 2022), and many more.

In this work, we aim to utilise KME to represent our invariant elementary distributions. Moreover, given new test samples, we aim to build a weighted ensemble of predictors from each elementary distributions, where the weights denote similarities between test-time domain and elementary distributions. However, this leads to two challenges, see Figure 2 for a visualisation.

First, we have no access to samples from the unknown elementary distributions. Thus, the elementary KME cannot be estimated directly from the samples at hand. To this end, we instead seek a proxy KME $\mu_{V_j} := N^{-1} \sum_{k=1}^{N} k(v_k^j, \cdot)$ for each elementary KME $\mu_{\mathbb{P}_j^e}$ using a domain basis $V_j$ where $V_j = \{v_1^j, \ldots, v_N^j\} \subseteq \mathcal{X}$ for each elementary domain $j \in \{1, \ldots, M\}$. Hence, the KME $\mu_{V_j}$ can be interpreted as the KME of the empirical probability measure $\hat{\mathbb{P}}_{V_j} = N^{-1} \sum_{k=1}^{N} \delta_{v_k^j}$. Here, we assume that $M = K$ meaning that we approximate all elementary distributions. The sets $V_j$ are referred to as *elementary domain basis*. Intuitively, the elementary domain basis $V_1, \ldots, V_M$ represents each elementary distribution by a set of vectors that mimic samples generated from the corresponding distribution.

The second challenge lies in measuring the similarity between samples at test-time and the elementary distributions. We consider two similarity measures utilising their RKHS presentations in the coming section.

With this two challenges tackled, we can proceed to build a convex combination of elementary domain-specific learning machine.

To this end, we instead seek a proxy KME $\mu_{V_j} := N^{-1} \sum_{k=1}^N k(v_k^j, \cdot)$ for each elementary KME $\mu_{\mathbb{P}_j^e}$ using a domain basis $V_j$ where $V_j = \{v_1^j, \ldots, v_N^j\} \subseteq \mathcal{X}$ for each elementary domain $j \in \{1, \ldots, M\}$.

## 4 Gated Domain Units

This section aims to answer the aforementioned question by instantiating the theoretical ideas presented in Section 3 as a neural network layer consists of a novel gated domain unit (GDU). For the purpose of implementation, let $x \in \mathbb{R}^{h \times w}$ denote the input data point and $h_\xi : \mathbb{R}^{h \times w} \to \mathbb{R}^e$ the feature extractor that transform the input into a representation $\tilde{x} \in \mathbb{R}^e$, e.g. ResNet-50 (He et al., 2016), DenseNet-121 Huang et al. (2017), Graph Isomorphism Networks with virtual nodes (GIN-virtual; Xu et al. (2019) and Gilmer et al. (2017)), DistillBERT Sanh et al. (2020). The final prediction layer is denoted as $g_\theta : \mathbb{R}^e \to \mathcal{Y}$.

GDU consists of three main components:

1. A similarity function $\gamma : \mathcal{H} \times \mathcal{H} \to \mathbb{R}$ defined on the RKHS that is shared across GDUs.

2. Elementary basis $V_j$, and

3. learning machine $f(\tilde{x}, \theta_j)$ for each elementary domain $j \in \{1, \ldots, M\}$.

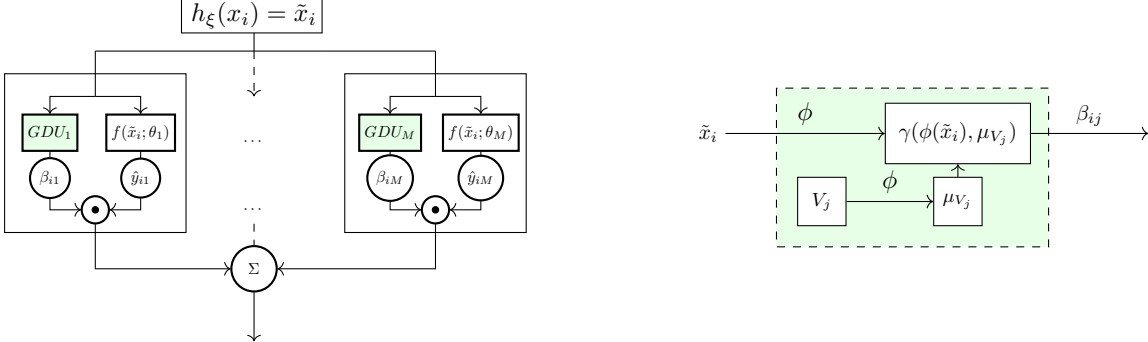

Figure 3: Visualization of the layer (upper panel) and its main component, the GDU (lower panel). The layer consists of multiple GDUs that represent the elementary distributions. During training, the GDUs learn the elementary domain basis $V_1, \ldots, V_M$ that approximate these distributions.

In Figure 3, we depict the architecture and how data is processed and information is stored. Given representation $\tilde{x}_i$, each GDU$_j$ computes the similarity $\beta_{ij}$ between this sample with their elementary basis $V_j$ using the similarity function $\gamma$ acting on their RKHS mappings, i.e. $\beta_{ij} = \gamma\left(\phi(\tilde{x}_i), \mu_{V_j}\right)$. Each GDU is then connected to a learning machine $f(\tilde{x}_i, \theta_j)$ that performs an elementary domain-specific inference. We build the final predictor by taking an ensemble of these learning machines by setting $g_\theta(\tilde{x}_i) = \sum_{j=1}^M \beta_{ij} f(\tilde{x}_i, \theta_j)$.

In summary, the GDUs leverage the invariant elementary distribution (I.E.D.) assumption and represent our algorithmic contribution. The elementary domain bases $\{V_j\}_j$ are stored as layer weight matrices, thus allowing us to learn them efficiently through backpropagation and avoid the dependency on problem-adaptive methods (e.g., domain-adversarial training) and domain information.

### 4.1 Choice of Similarity Functions

In this work, we consider two categories of similarity functions $\gamma$: one based on distances between inputs and the elementary basis, and the other on projection, akin to kernel sparse coding (Gao et al., 2010; 2013).

### 4.1.1 Geometry-based Generalization

We consider two geometry-based similarity measure for $\gamma$, namely the cosine similarity (CS) (Kim et al., 2019) and maximum mean discrepancy (MMD) (Borgwardt et al., 2006; Gretton et al., 2012). The former considers angles, while the latter considers distances to measure similarity. To ensure $\beta_{ij} := \gamma(\phi(\tilde{x}_i), \mu_{V_j})$ sum up to 1, we apply the kernel softmax function Gao et al. (2019):

$$\beta_{ij} = \gamma(\phi(\tilde{x}_i), \mu_{V_j}) = \frac{\exp\big(\kappa H(\phi(\tilde{x}_i), \mu_{V_j})\big)}{\sum_{k=1}^{M} \exp\big(\kappa H(\phi(\tilde{x}_i), \mu_{V_k})\big)},$$

where $\kappa > 0$ is a positive softness parameter for the kernel softmax. For CS, we pick

$$H(\phi(\tilde{x}_i), \mu_{V_j}) = \frac{\langle \phi(\tilde{x}_i), \mu_{V_j} \rangle_{\mathcal{H}}}{\|\phi(\tilde{x}_i)\|_{\mathcal{H}} \|\mu_{V_j}\|_{\mathcal{H}}}$$

while for MMD, we choose

$$H(\phi(\tilde{x}_i), \mu_{V_j}) = -\|\phi(\tilde{x}_i) - \mu_{V_j}\|_{\mathcal{H}}^2$$

instead. When given a batch of samples $\{\tilde{x}_l\}_{l=1}^{n_i}$, we can compute the similarity between the batch with each domain basis by replacing $\phi(\tilde{x}_i)$ with $\tilde{\mu}_i := \frac{1}{n_i} \sum \phi(\tilde{x}_l)$ in the above equations.

### 4.1.2 Projection-based Generalization

Besides geometry-based similarity, we can also seek for an optimal projection of $\phi(\tilde{x})$ onto the span of $\{\mu_{V_j}\}_{j=1}^{M}$ such that $\sum_{j=1}^{M} \|\phi(\tilde{x}_i) - \beta_{ij}\mu_{V_j}\|_{\mathcal{H}}^2$ is small. Note that as the subspace $\mathcal{H}_M := \text{span}\{\mu_{V_j} \mid j = 1, \ldots, M\}$ are learnt during training, we could incorporate pairwise orthogonality constraints, i.e. $\langle \mu_{V_j}, \mu_{V_i} \rangle = 0$ for any $i \neq j$, to maximises the "representation" power of this subspace. As a result, this leads to the best possible approximation of the projection of $\phi(\tilde{x}_i)$ onto $\mathcal{H}$ if $\phi(\tilde{x}_i)$ actually lies in the span. See Proposition 4.1.

In practice, given the $i^{th}$ batch of test samples $\{\tilde{x}_l\}_{l=1}^{n_i}$, we could instead find a projection that minimises $\sum_{j=1}^{M} \|\tilde{\mu}_i - \beta_{ij}\mu_{V_j}\|_{\mathcal{H}}^2$ where $\tilde{\mu}_i$ is the KME of the $i^{th}$ batch.

**Proposition 4.1.** *Given a distribution $\mathbb{P}$ where $\mu_{\mathbb{P}} \in \text{span}\{\mu_{V_j} \mid j = 1, ..., M\}$ such that $\langle \mu_{V_j}, \mu_{V_i} \rangle = 0$ for all $i \neq j$. The value of $\sum_{j=1}^{M} \|\mu_{\mathbb{P}} - \beta_j \mu_{V_j}\|_{\mathcal{H}}^2$ is minimised at $\beta_j^* = \langle \mu_{\mathbb{P}}, \mu_{V_j} \rangle_{\mathcal{H}} / \|\mu_{V_j}\|_{\mathcal{H}}^2$.*

See proof in Appendix A.2). In practice, when given a single sample $\tilde{x}_i$ we could set

$$\beta_{ij} = \frac{\langle \phi(\tilde{x}_i), \mu_{V_j} \rangle}{\|\mu_{V_j}\|_{\mathcal{H}}^2}.$$

If a batch of samples are given instead, we have,

$$\beta_{ij} = \frac{\langle \tilde{\mu}_i, \mu_{V_j} \rangle}{\|\mu_{V_j}\|_{\mathcal{H}}^2}.$$

### 4.2 Model Training

Following Zhuang et al. (2021), our learning objective function is formalized as $\mathcal{L}(g) + \lambda_D \Omega_D(\|g\|_{\mathcal{H}})$. The goal of the training can be described in terms of the two components of this function. Consider a batch of training data $\{x_1, \ldots, x_b\}$, where $b$ is the batch size. During training, we minimize the loss function

$$\mathcal{L}(g) = \frac{1}{b} \sum_{i=1}^{b} \mathcal{L}(\hat{y}_i, y_i) = \frac{1}{b} \sum_{i=1}^{b} \mathcal{L}(\sum_{j=1}^{M} \gamma(\phi(\tilde{x}_i), \mu_{V_j}) f_j(\tilde{x}_i), y_i)$$

for an underlying task and the respective batch size. In addition, our objective is that the model learns to distinguish between different domains. Thus, the regularization $\Omega_D$ is introduced to control the domain

basis. In our case, we require the regularization $\Omega_D$ to ensure that the KMEs of the elementary domain basis are able to represent the KMEs of the elementary domains. Therefore, we minimize the MMD between the feature mappings $\phi(\tilde{x}_i)$ and the associated representation $\sum_{j=1}^{M} \beta_{ij}\mu_{V_j}$. Note that $\beta_{ij} = \gamma(\phi(\tilde{x}_i), \mu_{V_j})$. Hence, the regularization $\Omega_D = \Omega_D^{OLS}$ is defined as

$$\Omega_D^{OLS}(\|g\|_{\mathcal{H}}) = \frac{1}{b}\sum_{i=1}^{b}\|\phi(\tilde{x}_i) - \sum_{j=1}^{M}\beta_{ij}\mu_{V_j}\|_{\mathcal{H}}^2.$$

The intuition is the objective to represent each feature mapping $\phi(\tilde{x}_i)$ by the domain KMEs $\mu_{V_j}$. Thus, we try to minimize the MMD between the feature map and a combination of $\mu_{V_j}$. The minimum of the stated regularization can be interpreted as the ordinary least square-solution of a regression-problem of $\phi(\tilde{x}_i)$ by the components of $\mathcal{H}_M$. In other words, we want to ensure that the basis $V_j$ is contained in feature mappings $\phi(\tilde{x}_i)$.

In the particular case of projection, we want the KME of the elementary domain to be orthogonal to ensure high expressive power. For this purpose, the additional term $\Omega_D^{\perp}$ will be introduced to ensure the desired orthogonality. Considering a kernel function with $k(x,x) = 1$, orthogonality would require the Gram matrix $K_{ij} = \langle \mu_{V_i}, \mu_{V_j} \rangle_{\mathcal{H}}$ to be close to the identity matrix $I$. There are a variety of methods for regularizing matrices available (Xie et al., 2017; Bansal et al., 2018). A well-known method to ensure orthogonality is the soft orthogonality regularization $\Omega_D^{\perp} = \lambda\|K - I\|_F^2$ (Bansal et al., 2018). As pointed out by Bansal et al. (2018), since spectral restricted isometry property and mutual coherence regularization can be a promising alternative, we provide an additional implementation for both. Hence, in the case of projection, the regularization is given by

$$\Omega_D(\|g\|_{\mathcal{H}}) = \lambda_{OLS}\Omega_D^{OLS}(\|g\|_{\mathcal{H}}) + \lambda_{ORTH}\Omega_D^{\perp}(\|g\|_{\mathcal{H}}), \; \lambda_{OLS}, \lambda_{ORTH} \geq 0.$$

Lastly, sparse coding is an efficient technique to find the least possible basis to recover the data subject to a reconstruction error (Olshausen & Field, 1997). Several such applications yield strong performances, for example in the field of computer vision (Lee et al., 2007; Yang et al., 2009). Kernel sparse coding transfers the reconstruction problem of sparse coding into $\mathcal{H}$ by using the mapping $\phi$, and, by applying a kernel function, the reconstruction error is quantified as the inner product (Gao et al., 2010; 2013). To ensure sparsity, we apply the $L_1$-norm on the coefficients $\beta$ and add $\Omega_D^{L_1}(\|\gamma\|) := \|\gamma(\phi(\tilde{x}_i), \mu_{V_j})\|_1$ to the regularization term $\Omega_D$ with the corresponding coefficient $\lambda_{L_1}$.

## 5 Experiments

First, we conduct a proof-of-concept study using a smaller Digits dataset in Section 5.1. We compare performance of the Gated Domain layer against a simple yet effective Empirical Risk Minimization (ERM) baseline. In addition, we compared our method against an ensemble of such ERM models, where the ensemble components are jointly trained on the combined source data. The number of ensembles in this experiment is equal to the number of source domains. We also use this digits dataset to discuss heuristic for choosing the main hyper-parameters and performing an ablation study. Second, in Section 5.2, we compare our approach to state-of-the-art DG methods, e.g., CORAL, IRM, and Group DRO, using a standardized WILDS benchmark Koh et al. (2021b). In our experiments, we distinguish two modes of training our layer: (i) **fine tuning**, where we extract features using a pre-trained model, and (ii) **end-to-end training**, where the feature extractor (FE) and the Gated Domain layer are jointly trained. Our code is publicly available for TensorFlow (https://github.com/im-ethz/pub-gdu4dg) and PyTorch (https://github.com/im-ethz/gdu4dg-pytorch).

### 5.1 Proof-of-concept based on Digits Classification

Following Feng et al. (2020) among others, we create a multi-source dataset by combining five publicly available digits image datasets, namely MNIST Lecun et al. (1998), MNIST-M Ganin & Lempitsky (2015), SVHN Netzer et al. (2011), USPS, and Synthetic Digits (SYN) Ganin & Lempitsky (2015). Each dataset, except USPS, is split into training and test sets of 25,000 and 9,000 images, respectively. For USPS, we take

the whole dataset for the experiment since it contains only 9,298 images. The task is to classify digits between zero and nine. We perform five leave-one-out experiments. For each experiment, one of these five datasets is considered an out-of-training target domain which is inaccessible during training, and the remaining four are constitute the source domains. Details regarding the model architecture, implementation as well as hyper-paremeters are given in Appendix B.1.

### 5.1.1 Parameter selection

From a practical perspective, our layer requires choosing two main hyper-parameters: the number of elementary domains $M$ and since we use the characteristics Gaussian kernel the corresponding parameter $\sigma$. The parameter $M$ defines the number of elementary domains and determines the size of the ensemble (i.e., the network size).

Here, we discuss a heuristic to set $M$. Similar to Matsuura & Harada (2020), we pass the training data through a FE and apply $k$-means clustering algorithm on the output. Then, we use the Davies-Bouldin score to select the optimal number of clusters as the basis to choose $M$. For example, using MNIST-M as the test domain, cluster scores suggest 10 to 12 clusters (see Figure 4).

As for the parameter $\sigma$, we resort to the median heuristic proposed in Muandet et al. (2016) that is $\sigma^2 = \mathrm{median}\{ \parallel \tilde{x}_i - \tilde{x}_j \parallel^2 : i, j = 1, \ldots, n\}$. While both heuristics require a pre-trained FE, cross-validation can act as a reasonable alternative given that we have access to adequate validation data (Koh et al., 2021a; Gulrajani & Lopez-Paz, 2020).

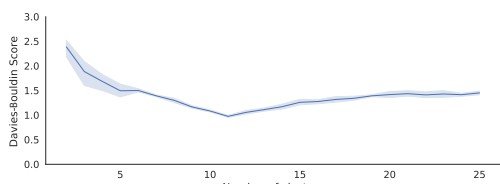

Figure 4: Visualization of the mean and standard deviation of the Davies-Bouldin Score across five runs for MNIST-M as the held-out test domain.

### 5.1.2 DG performance

Although the cluster score has its optimum around 12, we set $M$ to 10 for this experiment considering the additional computational complexity. The hyper-parameters relevant for our layer are summarized in Table 7 in Appendix B.1. In Table 2, we present the final results for our proof-of-concept experiment. We compare the performance of the Gated Domain layer with different similarity functions (CS, MMD, Projected) trained in fine tuning (FT) and end-to-end (E2E) modes. Additionally, we compare our approach against the Empirical Risk Minimization (ERM) algorithm, which trains the entire classification model using all available source data without enforcing any invariance. We also consider an ensemble of $M = 10$ ERM classifiers to make a fair comparison with the Gated Domain layers, which use multiple domains. Our method noticeably improves across all tasks the mean accuracy and decreases the standard deviation compared to the ERM and ERM ensemble baselines, making the results more stable across ten iterations.

Table 2: Results of the Digits experiment. All experiments were repeated ten times and the mean (standard deviation) accuracy is reported. Best mean results are **bold**.

|  |  | MNIST | MNIST-M | SVHN | USPS | SYN |
|---|---|---|---|---|---|---|
|  | *ERM* | *97.98 (0.34)* | *63.00 (3.20)* | *70.18 (2.74)* | *93.70 (1.74)* | *83.62 (1.47)* |
|  | *ERM ENSEMBLE* | *98.21 (0.39)* | *62.87 (1.50)* | *72.01 (3.59)* | *95.16 (0.89)* | *83.80 (1.22)* |
| **FT** | CS | **98.75 (0.09)** | 69.44 (0.57) | **79.65 (0.89)** | 96.43 (0.35) | **87.94 (0.61)** |
|  | MMD | 98.74 (0.12) | 69.32 (0.46) | 79.50 (1.19) | **96.54 (0.27)** | 87.78 (0.6) |
|  | PRO | 98.71 (0.11) | **69.44 (0.33)** | 79.4 (1.15) | 96.37 (0.36) | 87.68 (0.62) |
| **E2E** | CS | 98.50 (0.09) | 67.78 (2.25) | 76.76 (1.63) | 95.54 (0.45) | 87.41 (1.03) |
|  | MMD | 98.52 (0.11) | 68.80 (0.69) | 78.24 (1.09) | 95.22 (0.91) | 87.78 (0.94) |
|  | PRO | 98.13 (0.41) | 63.15 (4.32) | 75.04 (1.71) | 95.49 (0.79) | 86.04 (1.27) |

### 5.1.3 Ablation study

First, we discuss the effect of deviating from the cluster heuristic in choosing the number of elementary domains $M$. For this, we vary the corresponding parameter $M$ and analyze the classification performance to validate our heuristic and discover sensitivity of our Gated Domain layer to the choice of $M$. The results for the most challenging DG task (MNIST-M) are depicted in Figure 5 (left). In general, the performance difference is slight across the choice of $M$. However, for FT, we generally observe a higher classification performance than E2E, which also increases with higher $M$. For FT, we have the highest performance around the heuristically estimated $M$. In E2E, the performance is slightly lower, and we reach the highest performance when we set $M$ smaller than suggested by the heuristic. This behavior can be explained by the fact that in E2E, the feature extractor and GDUs are jointly trained, and the lower dimensional embedding (i.e., $\tilde{x}$) is stochastic, which makes learning to approximate the elementary distributions more challenging, especially when their number is high. These results support our heuristic, which we later apply to real-world datasets with a much larger number of source domains.

Lastly, we analyze the sensitivity of the GDUs to the number of basis vectors in each elementary domain ($N$). For this, we vary $N$ and keep the number of elementary domains fixed to the value suggested by the cluster heuristic. We observe that smaller $N$ yields higher results while having the beneficial side-effect of reducing the computational overhead. We explain such an effect on the performance with noise reduction and lowering variance as well as avoiding the curse of dimensionality, which makes the embedding sparse and dissimilar, thus degrading the performance of the similarity function. In other words, reducing the number of basis vectors can be viewed as a dimensionality reduction technique, similar to Principle Component Analysis for high-dimensional data.

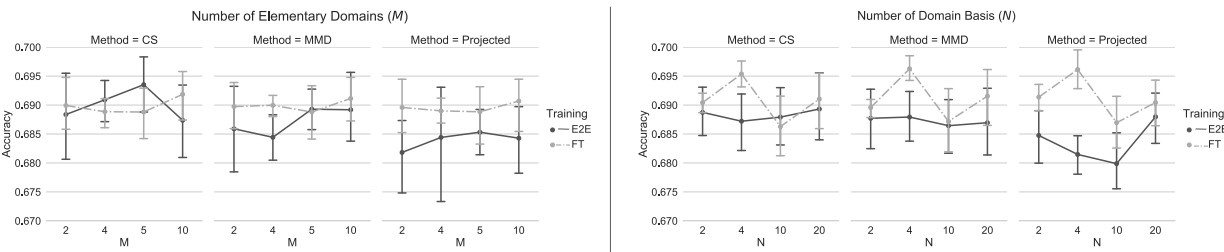

Figure 5: Mean and standard deviation of classification accuracy over 10 runs for varying number of elementary domains ($M$, left panel) and varying number of vector for each domain basis ($N$, right panel) for MNIST-M dataset.

The next paragraph focuses on ablating the regularization terms introduced in Section 4), which dependents on the form of generalization (i.e., geometry-based or projection-based). For the geometry-based generalization, the regularization is

$$\Omega_D\big(\|g\|_{\mathcal{H}}\big) = \lambda_{OLS}\Omega_D^{OLS}\big(\|g\|_{\mathcal{H}}\big) + \lambda_{L_1}\Omega_D^{L_1}(\|\gamma\|), \tag{1}$$

where $\lambda_{OLS}, \lambda_{L_1} \geq 0$. In the case of projection, the regularization is given by

$$\Omega_D\big(\|g\|_{\mathcal{H}}\big) = \lambda_{OLS}\Omega_D^{OLS}\big(\|g\|_{\mathcal{H}}\big) + \lambda_{ORTH}\Omega_D^{\perp}\big(\|g\|_{\mathcal{H}}\big) \tag{2}$$

with $\lambda_{OLS}, \lambda_{ORTH} \geq 0$.

We vary in Equation 1 and Equation 2 the corresponding weights $\lambda_{OLS}$ and $\lambda_{L_1}$ (Eq. 1) or $\lambda_{ORTH}$ (Eq. 2) in the interval of $[0; 0.1]$ and display the mean classification accuracy for the most challenging classification task of MNSIT-M in the form of a heatmap. In Figures 6 a-f, we see that the classification accuracy remains on an overall similar level which indicates that our method is not very sensitive to the hyper-parameter change for MNIST-M as the test domain. Nevertheless, we observe that ablating the regularization terms by setting the corresponding weights to zero decreases the classification results and the peaks in performance occur when the regularization is included during training of our method.

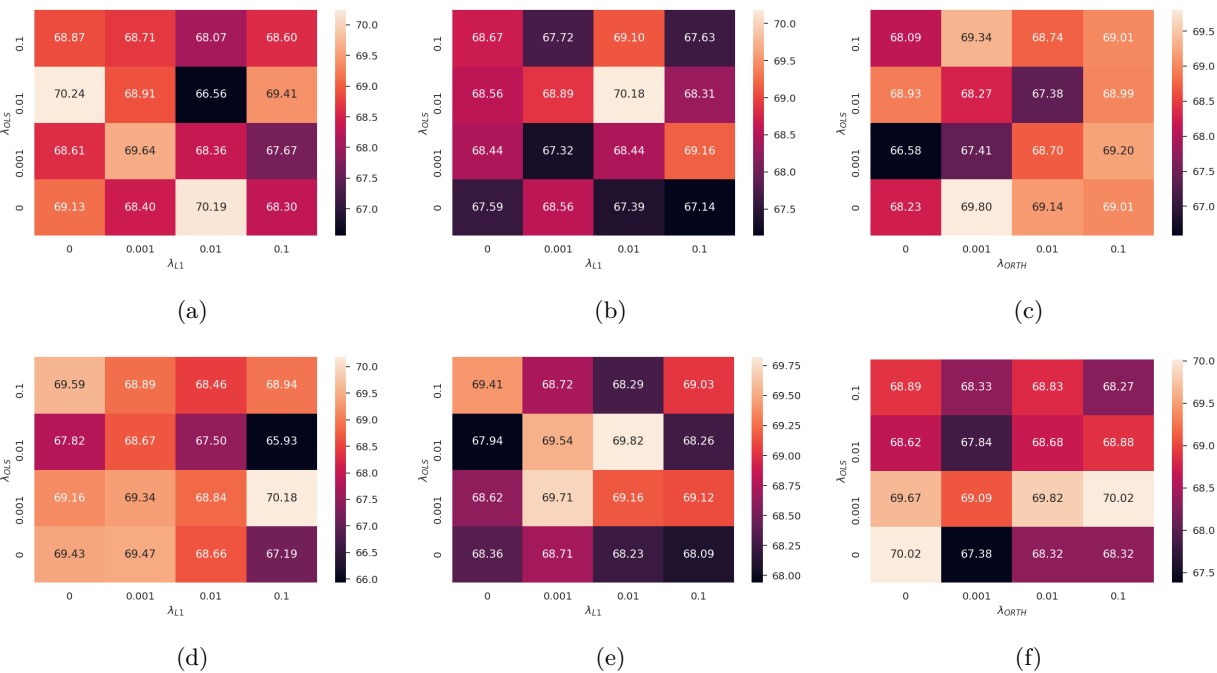

Figure 6: Classification results for varying $\lambda_{L_1}$ and $\lambda_{OLS}$ in the interval of $[0; 0.1]$ for FT-CS (a), E2E-CS (d), FT-MMD (b) and E2E-MMD (e) and for varying $\lambda_{ORTH}$ and $\lambda_{OLS}$ in the interval of $[0; 0.1]$ for FT-Projected (c) and E2E-Projected (f) Projection on MNIST-M.

Applying GDUs comes with additional overhead, especially the regularization term that ensures the orthogonality of the elementary domain bases. This additional effort raises a question whether ensuring the theoretical assumptions outweigh the much higher computational cost. Thus, in a second step, we analyze how the orthogonal regularization affects the orthogonality of the elementary domain bases (i.e., spectral restricted isometry property (SRIP) value) and the loss function (i.e., categorical cross-entropy).

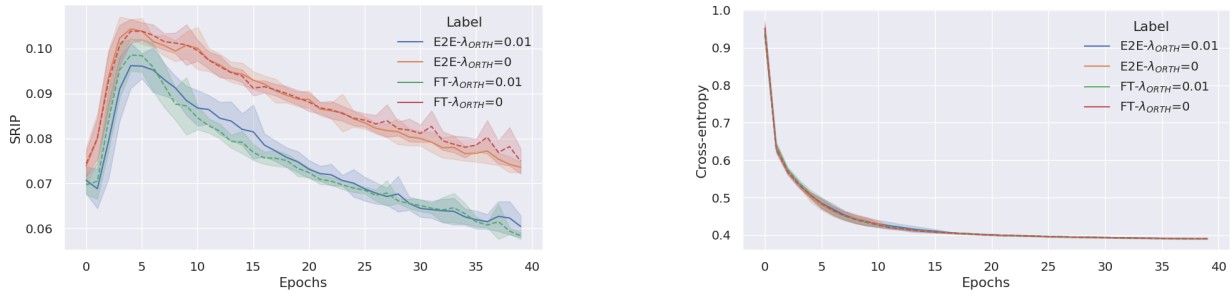

Figure 7: Effect of omitting the orthogonal regularization term $\Omega_D^\perp$. Spectral restricted isometry property (SRIP) (left) and categorical cross-entropy (right) with and without orthogonal regularization and their evolution during training for MNSIT-M dataset. The mean and standard deviation presented for End-to-end (E2E) and Fine-tuning (FT) training scenarios are calculated over 10 runs.

In Figure 7, we depict the mean and standard deviation of the spectral restricted isometry property (SRIP) value and loss over five runs for 40 epochs. The SRIP value can be tracked during training with our layer's callback functionalities. First, we observe that the elementary domains are almost orthogonal when initialized. Training the layer leads in the first epochs to a decrease in orthogonality. This initial decrease happens because cross-entropy has a stronger influence on the optimization than regularization in the first epochs. After five epochs, the cross-entropy decrease to a threshold when the regularization becomes more effective and the

orthogonality of the elementary domain bases increases again. In Figure 7, we also observe that ablating the orthogonal regularization, while leading to better orthogonality of the domains, does not significantly affect the overall cross-entropy during training.

This analysis revealed stable results across different sets of hyper-parameters. While the layer is not noticeably sensitive to the choice of regularization strength, we recommend not to omit the regularization completely, although the computational expenses decrease without the orthogonal regularization.

Third, in the E2E scenario, the GDUs are jointly trained with the feature extractor. In this particular scenario, we seek to understand how the GDU affect the latent representation learned by the feature extractor. For this, we project the output of the FE trained with a dense layer (ERM) and our layer by t-SNE in Figure 8. The GDU-trained FE yields more concentrated and bounded clusters in comparison to the one trained by ERM. Hence, we observe a positive effect on the FEs representation.

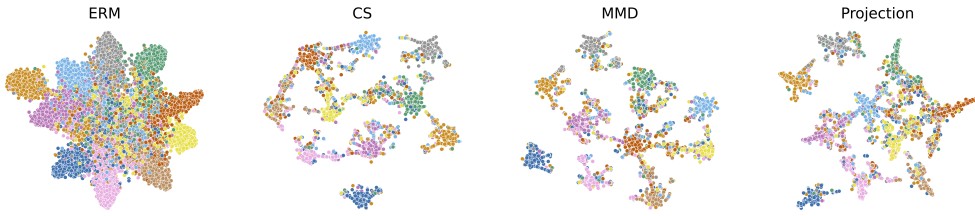

Figure 8: Visualization of t-SNE Embedding on unseen Synthetic Digits Dataset. Colors encode true label.

Lastly, one of the essential questions we seek to answer in this ablation study is on what distributional structure given by the dataset the learned elementary domains represent. Figure 9 depicts the t-SNE embedding of the joint source domains (left), the test, and the elementary domains learned by the GDUs (right). Nine of ten elementary domains form dense clusters around the joint source dataset, indicating that the GDUs learns to represent distributional structures in this multi-source dataset.

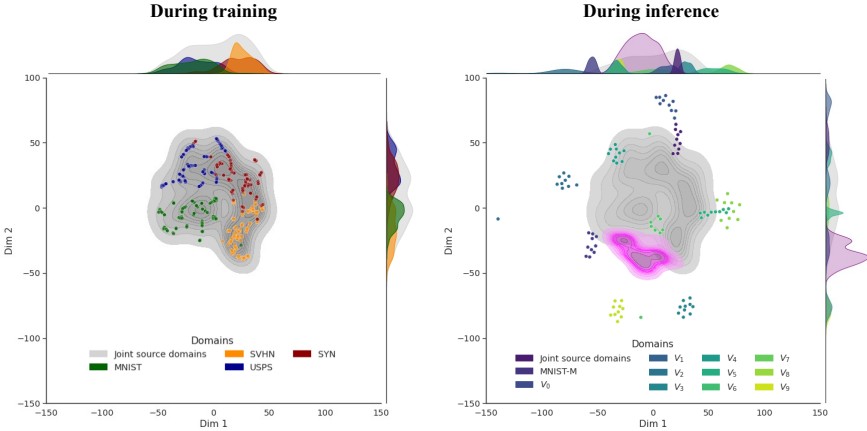

Figure 9: t-SNE visualization of source (left), the elementary and test domains (right). MNIST-M as held-out test domain, Gated Domain layer with MMD similarity trained in the fine tuning mode (FT-MMD). Colors encode source datasets (left) and elementary domains (right). Note: Marginals have different scales.

## 5.2 WILDS Benchmark

In addition to the ERM and ERM ensemble methods discussed in the previous section, we also compare our approach with state-of-the-art algorithms. Currently, two open-source benchmarks are available for a reproducible and rigorous comparison of DG methods, namely DomainBed Gulrajani & Lopez-Paz (2020) and WILDS (Koh et al., 2021a). WILDS is a semi-synthetic benchmark that operates under similar assumptions as the source component shift and represents DS occurring under real-world conditions. The advantage in

comparison to DomainBed is that WILDS reveals methods that work well on synthetic shifts may drastically fail on real-world shifts such as, for example, invariant risk minimization Gulrajani & Lopez-Paz (2020). To challenge the GDUs under potential model deployment, we chose WILDS.

The following methods reflect the potential and limitations of the research streams followed in DG (see Section 2.1), and thus, serve as our benchmark: CORAL, LISA, IRM, FISH, Group DRO, CGD, and ARM-BN. Appendix B.2.3 provides details on the baseline algorithms. Note that these algorithms require access to domain information, unlike the approaches discussed in the previous section. Depending on the nature of the data, these labels may be difficult to access, as discussed in Appendix B.2.2. We closely follow Koh et al. (2021a) for the experiments using eight datasets: *Camelyon17*, *FMoW*, *Amazon*, *iWildCam*, and *RxRx1*, *OGB-MolPCBA*, *Civil-Comments*, and *PovertyMap*. Details on datasets are given in Appendix B.2.1.

### 5.2.1 Benchmark results.

To provide a reasonable default parameter for our layer, we apply our heuristic to real-world conditions with a higher number of domains (e.g., 323 camera traps in iWildCam). Figure 10 depicts the cluster score for each of the eight dataset. The scores reach an agreement at $M = 5$, which we set as the default value considering the model complexity associated with a higher number of components. Table 8 from Appendix B.2 presents the default parameters of the Gated Domain layer used for the benchmarking. Please note that we used the same set of parameters for *all* WILDS experiments.

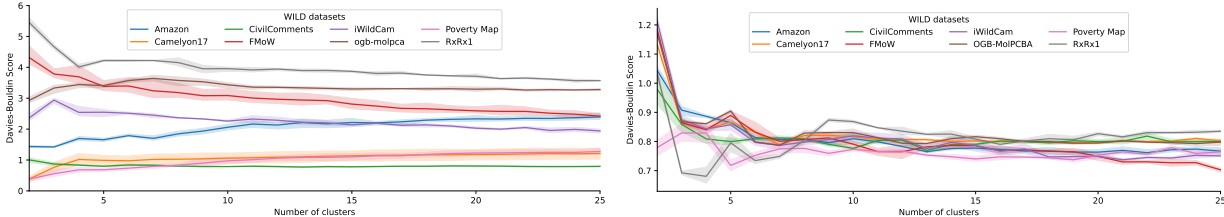

Figure 10: Visualization of the mean and standard deviation of Davies-Bouldin Score for each WILDS dataset with different number of clusters. The left panel depicts the scores when the output of the FE was clustered with *k-means*, while the right panel depicts the scores when the FE output was first embedded into a 2D space via t-SNE. Reducing dimensionality of the embeddings yielded better clustering results (i.e., lower Davies-Bouldin scores).

Table 3 reports the results for $M = 5$. Our approach achieves state-of-the-art OOD performance, however does not outperform the strongest benchmark LISA. Still, the consistent improvement to ERM is a significant advantage compared to the benchmarks since they fall at least once short against ERM (e.g., IRM in RxRx1, Fish in iWildCam, or Coral in OGB-MolPCBA). Thus, our method is especially appealing when domain information is challenging to obtain.

### 5.2.2 Worst-case vs. average-case performance.

Methods trained to perform well on the worst-case (e.g., underrepresented populations) are prone to a well-known trade-off, thus performing less well on the average cases Eastwood et al. (2022); Rice et al. (2021). Therefore, it is crucial to understand how DG methods are affected by this trade-off. We use the pre-defined average- and wors-case subsets to compute the performance scores and analyse how each method is affected based on the *Poverty*, *FMoW*, and *CivilComments* dataset.

As an example we provide the results for this analysis in Table 4. In general, we observe that this trade-off is apparent also for the benchmarking methods. Of particular note, our method balances this trade-off well, achieving improvements in worst- and average-case performance except for *CivilComments*. For this dataset, obtaining domain labels might be the most challenging, yet, using this information seems essential for generalization.

Table 3: Benchmarking results. A grey background highlights methods using no domain information for DG. We compute the metrics following Koh et al. (2021a) and report the mean (standard deviation). Best benchmark and GDU results are **bold**.

| | | Camelyon17 | FMoW | Amazon | iWildCam | RxRx1 | OGB-MolPCBA | Civil Comments | Poverty Map |
|---|---|---|---|---|---|---|---|---|---|
| **No. domains** | | 5 | 16 x 5 | 3,920 | 323 | 51 | 120,084 | 16 | 23 x 2 |
| **Scores** | | Avg Acc | Worst-Reg Acc | 10% Acc | Macro F1 | Avg Acc | Avg Worst-Region Acc | Worst-Group Acc | Worst-U/R R |
| **ERM** | | 70.3 (6.4) | 31.3 (0.17) | 53.8 (0.8) | 31.0 (1.3) | 29.9 (0.4) | **27.2 (0.3)** | 56.0 (3.6) | 0.45 (0.06) |
| **ERM Ensemble** | | 70.0 (9.4) | 34.3 (0.18) | 54.0 (0.5) | 29.5 (0.4) | 29.6 (0.3) | 26.9 (0.3) | 55.6 (0.8) | **0.52 (0.08)** |
| **CORAL** | | 59.3 (7.7) | 32.8 (0.66) | 52.9 (0.8) | **32.8 (0.1)** | 28.4 (0.3) | 17.9 (0.5) | 65.6 (1.3) | 0.44 (0.07) |
| **Fish** | | 74.7 (7.1) | **34.6 (0.18)** | 53.3 (0.0) | 22.0 (1.8) | - | - | **75.3 (0.6)** | - |
| **IRM** | | 64.2 (8.1) | 32.8 (2.09) | 52.4 (0.8) | 15.1 (4.9) | 8.2 (1.1) | 15.6 (0.3) | 64.2 (8.1) | 0.43 (0.07) |
| **Group DRO** | | 68.4 (7.3) | 31.1 (1.66) | 53.3 (0.0) | 23.9 (2.1) | 22.5 (0.3) | 22.4 (0.6) | 68.4 (7.3) | 0.39 (0.06) |
| **LISA** | | **77.1 (6.9)** | 35.5 (0.81) | **54.7 (0.0)** | - | **31.9 (1.0)** | - | 72.9 (1.0) | - |
| **CGD** | | 69.4 (7.9) | 32.0 (2.26) | - | - | - | - | 69.1 (1.9) | 0.43 (0.04) |
| **ARM-BN** | | - | 24.4 (0.54) | - | 23.3 (2.8) | 31.2 (0.1) | - | - | - |
| | | | | | *Ours* | | | | |
| **FT** | CS | 68.5 (8.3) | 31.8 (1.2) | **54.2 (0.8)** | **31.2 (0.8)** | **29.9 (0.3)** | **27.5 (0.2)** | 56.0 (3.7) | 0.62 (0.13) |
| | MMD | 67.9 (8.0) | 31.9 (1.2) | **54.2 (0.8)** | **31.2 (0.8)** | **29.9 (0.3)** | **27.5 (0.2)** | 55.9 (3.7) | 0.62 (0.13) |
| | PRO | 66.7 (8.9) | 31.8 (1.0) | **54.2 (0.8)** | 30.4 (1.0) | 29.8 (0.3) | 27.4 (0.2) | 56.5 (2.9) | **0.62 (0.12)** |
| **E2E** | CS | 66.7 (8.9) | 34.0 (1.9) | 53.9 (0.7) | 27.8 (2.1) | 29.7 (0.4) | 26.9 (0.1) | 55.9 (0.8) | 0.46 (0.07) |
| | MMD | 65.7 (6.7) | **34.4 (0.7)** | **54.2 (0.8)** | 27.4 (1.6) | 29.6 (0.2) | 27.0 (0.5) | 55.8 (0.7) | 0.50 (0.06) |
| | PRO | **72.3 (9.5)** | 32.9 (0.8) | 53.8 (0.8) | 30.1 (1.2) | 29.0 (0.2) | 26.6 (0.3) | **56.4 (2.1)** | 0.49 (0.07) |

Table 4: Detailed results on the trade-off between worst- and average case performance. A grey background highlights methods using no domain information for DG. We compute the metrics following Koh et al. (2021a) and report the mean (standard deviation). Best benchmark and GDU results are **bold**.

| | | PovertyMap | | CivilComments | | FMoW | |
|---|---|---|---|---|---|---|---|
| | | Average-case | Worst-case | Average-case | Worst-case | Average-case | Worst-case |
| | | Pearson r | worst-U/R Pearson r | Avg Acc | Worst-region Acc | Avg Acc | Worst-region Acc |
| **ERM** | | 0.78 (0.04) | 0.45 (0.06) | **92.1 (0.1)** | 56.0 (3.6) | 52.7 (0.23) | 31.3 (0.17) |
| **ERM Ensemble** | | **0.8 (0.04)** | **0.52 (0.08)** | **92.1 (0.0)** | 55.6 (0.8) | **53.6 (0.06)** | 34.3 (1.85) |
| **CORAL** | | 0.78 (0.05) | 0.44 (0.07) | 88.7 (0.5) | 65.6 (1.3) | 50.1 (0.07) | 32.8 (0.66) |
| **Fish** | | - | - | 89.3 (0.3) | **75.3 (0.6)** | 51.8 (0.32) | 34.6 (0.18) |
| **IRM** | | 0.77 (0.05) | 0.43 (0.07) | 88.8 (0.7) | 66.3 (2.1) | 50.4 (0.75) | 32.8 (2.09) |
| **Group DRO** | | 0.75 (0.07) | 0.39 (0.06) | 89.9 (0.5) | 70.0 (2.0) | 52.8 (1.15) | 31.1 (1.66) |
| **LISA** | | - | - | 90.1 (0.3) | 72.9 (1.0) | 52.8 (1.15) | 35.5 (0.81) |
| **CGD** | | 0.77 (0.04) | 0.43 (0.04) | 89.6 (0.4) | 69.1 (1.9) | 50.6 (1.39) | 32.0 (2.26) |
| **ARM-BN** | | - | - | - | - | 23.8 (2.0) | **72.7 (2.0)** |
| **FT** | CS | **0.85 (0.04)** | **0.62 (0.13)** | **92.3 (0.2)** | 56.0 (3.7) | **53.1 (0.22)** | 31.8 (1.24) |
| | MMD | **0.85 (0.04)** | **0.62 (0.13)** | **92.3 (0.2)** | 56.0 (3.7) | **53.1 (0.22)** | 31.9 (1.17) |
| | PRO | 0.84 (0.04) | **0.62 (0.12)** | **92.3 (0.3)** | 56.5 (2.9) | 52.7 (0.14) | 31.8 (1.08) |
| **E2E** | CS | 0.78 (0.06) | 0.46 (0.07) | **92.3 (0.2)** | 55.9 (0.8) | 53.4 (0.25) | **34.4 (1.86)** |
| | MMD | 0.80 (0.05) | 0.50 (0.06) | **92.3 (0.2)** | 55.9 (0.7) | 52.7 (0.45) | **34.4 (0.71)** |
| | PRO | 0.78 (0.04) | 0.49 (0.07) | 92.2 (0.3) | **56.4 (2.1)** | 52.7 (0.68) | 32.9 (0.78) |

### 5.2.3 IID vs. OOD performance.

The relationship between IID and OOD performance can vary, from a positive correlation to a trade-off Wenzel et al. (2022); Teney et al. (2022). Therefore, as a last step, we seek to understand how our method performs on IID data compared to the OOD data, and compare the performance to the benchmarking methods. For this analysis, we leverage the IID and OOD subsets of the *iWildCam* and *RxRx1* experiments.

The IID and OOD performances are presented in Table 5. The trade-off between OOD and IID performance is apparent for some methods (e.g., ARM-BN in *RxRx1*, Coral in *iWildCam*). For *RxRx1*, the GDUs' OOD performance equals that of ERM; however, we improve in IID performance. For *iWildCam*, we observe a positive relation between IID and OOD performance. In summary, our modular layer consisting of the GDUs balance the performance on the IID and OOD subsets.

Table 5: Detailed results on the trade-off between IID and OOD performance. A grey background highlights methods using no domain information for DG. We compute the metrics following Koh et al. (2021a) and report the mean (standard deviation). Best benchmark and GDU results are **bold**.

| | | RxRx1 | | iWildCam | | | |
| | | IID | OOD | IID | | OOD | |
| | | ACCURACY | ACCURACY | MACRO F1 SCORE | AVG ACC | MACRO F1 SCORE | AVG ACC |
|---|---|---|---|---|---|---|---|
| ERM | | **35.9 (0.4)** | 29.9 (0.4) | **47.1 (1.5)** | **75.7 (0.4)** | 30.8 (1.3) | 71.5 (2.6) |
| ERM Ensemble | | 35.6 (0.4) | 29.9 (0.4) | 45.3 (2.1) | 74.6 (0.2) | 29.5 (0.4) | 69.6 (3.2) |
| CORAL | | 34.0 (0.3) | 28.4 (0.3) | 43.6 (3.3) | 73.8 (0.3) | **32.7 (0.2)** | **73.3 (4.3)** |
| Fish | | - | - | 40.3 (0.6) | 73.8 (0.1) | 22.0 (1.8) | 64.7 (2.6) |
| IRM | | 9.9 (1.4) | 8.2 (1.1) | 22.4 (7.7) | 59.8 (8.2) | 15.1 (4.9) | 59.7 (3.8) |
| Group DRO | | 28.1 (0.3) | 22.5 (0.3) | 37.5 (1.9) | 71.6 (2.7) | 23.8 (2.0) | 72.7 (2.0) |
| LISA | | 41.1 (1.3) | **31.9 (1.0)** | - | - | - | - |
| CGD | | - | - | - | - | - | - |
| ARM-BN | | 34.9 (0.2) | **31.2 (0.1)** | 27.5 (5.4) | 62.0 (4.0) | 23.3 (2.8) | 70.2 (2.4) |
| | CS | 36.0 (0.4) | 29.7 (0.4) | 47.7 (0.1) | **76.6 (0.3)** | **31.2 (0.8)** | **72.7 (1.8)** |
| FT | MMD | 36.0 (0.2) | 29.6 (0.2) | 47.7 (0.1) | **76.6 (0.3)** | **31.2 (0.8)** | **72.7 (1.8)** |
| | PRO | 35.1 (0.3) | 29.0 (0.2) | **48.2 (0.5)** | 76.3 (0.4) | 30.5 (1.0) | 72.5 (1.8) |
| | CS | **36.2 (0.4)** | **29.9 (0.3)** | 44.2 (2.0) | 74.9 (0.5) | 27.8 (2.2) | 69.7 (0.3) |
| E2E | MMD | **36.2 (0.4)** | **29.9 (0.3)** | 43.4 (1.6) | 75.3 (0.2) | 27.4 (1.6) | 68.3 (2.6) |
| | PRO | 36.0 (0.5) | 29.8 (0.3) | 45.4 (3.2) | 75.0 (1.4) | 30.1 (1.3) | 71.6 (4.7) |

## 6 Discussion and Conclusion

This work investigated the domain generalization (DG) problem under the I.E.D. assumption: *real-world distributions are composed of elementary distributions that remain invariant across different domains*. We showed its theoretical implications and empirical effectiveness when instantiated in the form of a neural architecture called the GDUs.

**Limitations.** In contrast to prior work, mostly focusing on aligning training source domains on an observational level given fixed domain labels, our assumption enables modeling the lower-level shifts between them. However, the number of elementary distributions $M$ must be specified in advance by the practitioners. While the application-agnostic heuristic used in our experiments has proven effective, the optimal choice of $M$ remains an open problem. Furthermore, our GDU layer induces additional computational overhead due to the regularization and model size that increases as a function of the number of elementary domains. Noteworthy, our improvement is achieved with a relatively small number of elementary domains indicating that the increased complexity is not a coercive consequence of applying the GDUs. Also, the results achieved are not a consequence of increased complexity, as the ensemble baseline shows.

**Outlook.** Finally, for the I.E.D. assumption and GDUs to be fully adopted, yielding novel applications that tackle DG in general, it is essential to develop flexible architectures that can efficiently adapt the number of GDUs during training. For example, one could increase $M$ adaptively during training given proper clustering criteria. Following the deterministic annealing formulation Rose (1998), one could split an elementary domain into two once a variance within the domain becomes too large. A generalization of the I.E.D. assumption to situations when some elementary distributions can differ across domains is also an interesting direction to pursue.

## Acknowledgments

This project was partially funded by a grant provided by the Hasler Stiftung (Project No. 22079) and the Swiss Heart Failure Network, PHRT122/2018DRI14 (J. M. Buhmann, PI). Simon Föll acknowledges the Helmholtz Information & Data Science Academy (HIDA) for providing financial support enabling a short-term research stay at CISPA (Application No. 14877). We also thank András Sass for contributing to the PyTorch implementation.

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

# A Proofs

## A.1 Proof of Proposition 3.1

*Proof.* The result holds trivially for $K = 1$. For $K \geq 2$ and by the I.E.D assumption, $\mathbb{P}^s(X,Y) = \sum_{j=1}^{K} \alpha_j \mathbb{P}_j(X,Y)$ for some $\boldsymbol{\alpha} \in \Delta^K$. Then, we can write the risk functional for each $f \in \mathcal{F}$ as $R(f) = \int \mathcal{L}(y, f(x)) \, d\mathbb{P}^s(x,y) = \int \mathcal{L}(y, f(x)) \, d(\sum_{j=1}^{K} \alpha_j \mathbb{P}_j(x,y)) = \sum_{j=1}^{K} \alpha_j \int \mathcal{L}(y, f(x)) \, d\mathbb{P}_j(x,y) = \sum_{j=1}^{K} \alpha_j R_j(f)$ where $R_j : \mathcal{F} \to \mathbb{R}_+$ is the elementary risk functional associated with the elementary distribution $\mathbb{P}_j(X,Y)$. Hence, the Bayes predictors satisfy

$$f^* \in \arg\min_{f \in \mathcal{F}} R(f) = \arg\min_{f \in \mathcal{F}} \sum_{j=1}^{K} \alpha_j R_j(f). \tag{A.3}$$

Since the rhs of equation A.3 corresponds to the linear scalarization of a multi-objective function $(R_1, \ldots, R_K)$, its solution (i.e., a stationary point) is Pareto-optimal with respect to these objective functions (Ma et al., 2020, Definition 3.1); see, also, (Hillermeier, 2001a;b). That is, the Bayes predictors for the data distribution that satisfies the I.E.D assumption must belong to the set $\{f^* : f^* = \arg\min_{f \in \mathcal{F}} \sum_{j=1}^{K} \alpha_j R_j(f), \ \boldsymbol{\alpha} \in \Delta^K\} \subset \mathcal{F}_{\text{Pareto}} \subset \mathcal{F}$, where $\mathcal{F}_{\text{Pareto}}$ comprises predictors that are Pareto-optimal with respect to a hypothesis space $\mathcal{F}$ and risk functionals $(R_1, \ldots, R_K)$. $\qquad\square$

## A.2 Proof of Proposition 4.1

*Proof.* Suppose we have a representation,

$$\mu_{\mathbb{P}} = \sum_{j=1}^{M} \beta_j \mu_{V_j} \qquad \langle \mu_{V_i}, \mu_{V_i} \rangle_{\mathcal{H}} = 0 \, \forall i \neq j, \tag{A.1}$$

i.e. $\{\mu_{V_1}, \ldots, \mu_{V_m}\}$ are pairwise orthogonal. We want to minimize the MMD by minimizing

$$\left\| \mu_{\mathbb{P}} - \sum_{j=1}^{M} \beta_j \mu_{V_j} \right\|_{\mathcal{H}}^2 = \underbrace{\langle \mu_{\mathbb{P}}, \mu_{\mathbb{P}} \rangle_{\mathcal{H}}}_{\|\mu_{\mathbb{P}}\|_{\mathcal{H}}^2 =} - 2\langle \mu_{\mathbb{P}}, \sum_{j=1}^{M} \beta_j \mu_{V_j} \rangle_{\mathcal{H}} + \langle \sum_{i=1}^{M} \beta_i \mu_{V_i}, \sum_{j=1}^{M} \beta_j \mu_{V_j} \rangle_{\mathcal{H}} \tag{A.2}$$

$$= \|\mu_{\mathbb{P}}\|_{\mathcal{H}}^2 - 2\sum_{j=1}^{M} \beta_j \langle \mu_{\mathbb{P}}, \mu_{V_j} \rangle_{\mathcal{H}} + \sum_{i=1}^{M} \sum_{j=1}^{M} \beta_i \beta_j \underbrace{\langle \mu_{V_i}, \mu_{V_j} \rangle_{\mathcal{H}}}_{\delta_{ij} \langle \mu_{V_i}, \mu_{V_j} \rangle_{\mathcal{H}} =} \tag{A.3}$$

$$= \|\mu_{\mathbb{P}}\|_{\mathcal{H}}^2 - 2\sum_{j=1}^{M} \beta_j \langle \mu_{\mathbb{P}}, \mu_{V_j} \rangle_{\mathcal{H}} + \sum_{j=1}^{M} \beta_j^2 \|\mu_{V_j}\|_{\mathcal{H}}^2 . \tag{A.4}$$

By defining

$$\Phi(\beta) := \left\| \mu_{\mathbb{P}} - \sum_{j=1}^{M} \beta_j \mu_{V_j} \right\|_{\mathcal{H}}^2 , \tag{A.5}$$

we can simply find the optimal $\beta_j$ by using the partial derivative

$$\frac{\partial \Phi}{\partial \beta_j} = -2\langle \mu_{\mathbb{P}}, \mu_{V_j} \rangle_{\mathcal{H}} + 2\beta_j \|\mu_{V_j}\|_{\mathcal{H}}^2 \overset{!}{=} 0 \tag{A.4}$$

$$\Leftrightarrow \beta_j \|\mu_{V_j}\|_{\mathcal{H}}^2 = \langle \mu_{\mathbb{P}}, \mu_{V_j} \rangle_{\mathcal{H}} \tag{A.5}$$

$$\Leftrightarrow \beta_j^* = \frac{\langle \mu_{\mathbb{P}}, \mu_{V_j} \rangle_{\mathcal{H}}}{\|\mu_{V_j}\|_{\mathcal{H}}^2} . \tag{A.6}$$

Please note that the function $\Phi$ is convex. $\qquad\square$

## B  Experiments

In this section, we provide a detailed description of the DG experiment presented in Section 5. Our Digits experiments are implemented using TensorFlow 2.4.1 and TensorFlow Probability 0.12.1. For the WILDS benchmarking we use our PyTorch (version 1.11.0). All source code is available on GitHub `https://github.com/im-ethz/pub-gdu4dg` (TensorFlow) and `https://github.com/im-ethz/gdu4dg-pytorch` (PyTorch).

For the Gated Domain layer, we considered two modes of model training: fine tuning (FT) and end-to-end training (E2E). In FT scenario, we first pre-train the FE in the ERM single fashion. Then, we extract features using the pre-trained model and pass them to the Gated Domain layer for training the latter. For the E2E training, however, the whole model including the FE and Gated Domain layer is trained jointly from the very beginning.

### B.1  Digits Experiment

Our experiment setup is closely related to Peng et al. (2019); Feng et al. (2020); Zhang et al. (2020); Zhao et al. (2018). We used the digits data from `https://github.com/FengHZ/KD3A` [last accessed on 2022-05-17, available under MIT License.] published in Feng et al. (2020). Our experimental setup regarding datasets, data loader, and FE are based on existing work (Feng et al., 2020; Peng et al., 2019). The structure of the FE is summarized in Table 6 and the subsequent learning machine is a dense layer.

In the Empirical Risk Minimization (ERM) single experiment, we add a dense layer with 10 outputs (activation=$tanh$) as a classifier to the FE. In the Empirical Risk Minimization (ERM) ensemble experiment, we add $M$ classification heads (a dense layers with 10 outputs and $tanh$ activation each) to the FE and average their output for the final prediction. This sets a baseline for our Gated Domain layer to show performance gain against the ERM model with the same number of learning machines.

Table 6: Feature Extractor used for the Digits Experiment

| FEATURE EXTRACTOR | |
| --- | --- |
| LAYER TYPE | OUTPUT SHAPE |
| 2D-CONVOLUTIONAL LAYER | (32, 32, 64) |
| BATCH NORMALIZATION | (32, 32, 64) |
| MAXPOOLING 2D | (16, 16, 64) |
| 2D-CONVOLUTIONAL LAYER | (16, 16, 64) |
| BATCH NORMALIZATION | (16, 16, 64) |
| MAXPOOLING 2D | (8, 8, 64) |
| 2D-CONVOLUTIONAL LAYER | (8, 8, 128) |
| BATCH NORMALIZATION | (8, 8, 128) |
| MAXPOOLING 2D | (4, 4, 128) |
| FLATTEN | (2048) |
| DENSE LAYER | (3072) |
| BATCH NORMALIZATION | (3072) |
| DROPOUT | (3072) |
| BATCH NORMALIZATION | (2048) |
| DENSE LAYER | (2048) |

For training, we resorted to the Adam optimizer with a learning rate of 0.001. We used early stopping and selected the best model weights according to the validation accuracy. For the validation data, we used the combined test splits only of the respective source datasets. The batch size was set to 512. Although the Gated Domain layer requires more computation resources than the ERM models, all digits experiments were conducted on a single GPU (NVIDIA GeForce RTX 3090).

Table 7: Parameters for Gated Domain layer in Digits and Digit-DG Experiments for the Fine Tuning (FT) and End-to-end training (E2E) Settings. In case of Projection, we chose the spectral restricted isometry property (SRIP) as the orthogonal regularization $\Omega_D^\perp$. In the FT setting, we applied the median heuristics presented above to estimate $\sigma$ of the Gaussian kernel function, where the estimator is denoted as $\hat{\sigma}$. Since median heuristic is not applicable for the E2E scenario, $\sigma$ was fixed to 7.5 for E2E.

| EXPERIMENT | | M | N | $\lambda_{L_1}$ | $\lambda_{OLS}$ | $\lambda_{ORTH}$ | $\sigma$ | $\kappa$ |
| --- | --- | --- | --- | --- | --- | --- | --- | --- |
| | CS | 5 | 10 | $1e^{-3}$ | $1e^{-3}$ | - | $\hat{\sigma}$ | 2 |
| FT | MMD | 5 | 10 | $1e^{-3}$ | $1e^{-3}$ | - | $\hat{\sigma}$ | 2 |
| | PROJECTION | 5 | 10 | $1e^{-3}$ | $1e^{-3}$ | $1e^{-8}$ | $\hat{\sigma}$ | - |
| | CS | 5 | 10 | $1e^{-3}$ | $1e^{-3}$ | - | 7.5 | 2 |
| E2E | MMD | 5 | 10 | $1e^{-3}$ | $1e^{-3}$ | - | 7.5 | 2 |
| | PROJECTION | 5 | 10 | $1e^{-3}$ | $1e^{-3}$ | $1e^{-8}$ | 7.5 | - |

## B.2 WILDS Benchmarking Experiments

For comparison of our approach and benchmarking, we followed the standard procedure of WILDS experiments, described in Koh et al. (2021a). As a technical note, all WILDS experiments have been implemented in Pytorch (version >= 1.7.0) based on the specifications made in Koh et al. (2021a) and their code published on `https://github.com/p-lambda/wilds` [last accessed on 2022-05-17, available under MIT License]. Table 8 presents the parameters of the Gated Domain layer used for *all* WILDS experiments. The results for the benchmarks were retrieved from the official leaderboard `https://wilds.stanford.edu/leaderboard/` [last accessed on 2022-09-26].

Table 8: Parameters for Gated Domain layer in WILDS experiments for the Fine Tuning (FT) and End-to-end training (E2E) Settings.

| | EXPERIMENT | | M | N | $\lambda_{L_1}$ | $\lambda_{OLS}$ | $\lambda_{ORTH}$ | $\sigma$ | $\kappa$ |
|---|---|---|---|---|---|---|---|---|---|
| WILDS BENCHMARK | FT AND E2E | CS | 5 | 10 | $1e^{-3}$ | $1e^{-3}$ | - | 4 | 2 |
| | | MMD | 5 | 10 | $1e^{-3}$ | $1e^{-3}$ | - | 4 | 2 |
| | | PROJECTION | 5 | 10 | - | $1e^{-3}$ | $1e^{-3}$ | 16 | - |

### B.2.1 Benchmark datasets

**Camelyon17** In medical applications, the goal is to apply models trained on a comparatively small set of hospitals to a larger number of hospitals. For this application, we study images of tissue slides under a microscope to determine whether a patient has cancer or not. Shifts in patient populations, slide staining, and image acquisition can impede model accuracy in previously unseen hospitals. Camelyon17 comprises images of tissue patches from five different hospitals. While the first three hospitals are the source domains (302,436 examples), the forth and fifth are the validation (34,904 examples) and test domain (85,054 examples), respectively.

We strictly follow the specifications made in (Koh et al., 2021a) using DenseNet-121 (Huang et al., 2017) as the feature extractor, a learning rate of 10e-3, an $L_2$ regularization of 10e-2, a batch size of 32, and stochastic gradient with momentum of 0.9. We trained for 5 epochs using early stopping.

We deviate from the specifications made in (Koh et al., 2021a) in terms of the FE. We use the FE from Feng et al. (2020); Peng et al. (2019) since we observed a higher mean accuracy and faster training than with the by Koh et al. (2021a) originally proposed DenseNet-121 FE (Huang et al., 2017). We trained the FE from scratch. Both, ERM and the DG were trained over 250 epochs with early stopping, a learning rate of 0.001, which is reduced by a factor of 0.2 if the cross-entropy loss has not improved after 10 epochs. All results were aggregated over ten runs.

**FMoW** Analyzing satellite images with machine learning (ML) models may enable novel possibilities in tackling global sustainability and economic challenges such as population density mapping and deforestation tracking. However, satellite imagery changes over time due to human behavior (e.g., infrastructure development), and the extent of change is different in each region. The Functional Map of the World (FMoW) dataset consists of satellite images from different continents and years: training (76,863 images; between 2002–2013), validation (19,915 images; between 2013 and 2016), and test (22,108 images, between 2016–2017). The objective is to determine one of 62 building types (e.g., shopping malls) and land-use.

As instructed in Koh et al. (2021a), we used the DenseNet-121 (Huang et al., 2017) pre-trained on ImageNet without L2-regularization. For the optimization, we use the Adam optimizer with a learning rate of 1e-4, which is decayed by a factor of 0.96 per epoch. The models were trained for 50 epochs with early stopping and a batch size of 64. Additionally, we report the worst-region accuracy, which is a specific metric used for FMoW. This worst-region accuracy reports the worst accuracy across the following regions: Asia, Europe, Africa, America, and Oceania (see Koh et al. (2021a) for the details). Again, we report the results over three runs.

**Amazon**   Recent research shows that consumer-facing machine learning application large performance disparities across different set of users. To study this performance disparities, WILDS (Koh et al., 2021a) leverages a variant of the Amazon Review dataset. The Aamazon-WILDS dataset is composed of data from 3,920 domains (number of reviewers) and the task is a multi-class sentiment classification, where the model receives a review text and has to predict the rating from one to five. To split this dataset, a between training, validation, and test disjoint set of reviewers is used: training (245,502 reviews from 1,252 reviewers), validation (100,050 reviews from 1,334 reviewers), test (100,050 reviews from 1,334 reviewers).

For the experiments and baseline models, we use the specifications made in Koh et al. (2021a). As for the FE, we used DistilBERT-base-uncased models (Sanh et al., 2020). For ERM, we use a batch size of 8, learning rate 1e-5, L2 regularization of 0.01, 3 epochs with early stopping and 512 as the maximum length of tokens. For training the Gated Domain layer, we used the same specifications as made for ERM. The performance is measured in 10th percentile accuracy.

**iWildsCam**   Wildlife camera traps offer an excellent possibility to understand and monitor biodiversity loss. However, images from different camera traps differ in illumination, color, camera angle, background, vegetation, and relative animal frequencies. We use the iWildsCam dataset consisting of 323 different camera traps positioned in different locations worldwide. In the dataset, we refer to different locations of camera traps as different domains, in particular 243 training traps (129,809 images), 32 validation traps (14,961 images), and 48 test traps (42,791 images). The objective is to classify one of 182 animal species.

Following the instructions by Koh et al. (2021a), we used again the ResNet50 pre-trained on ImagNet (He et al., 2016). For ERM, we used a learning rate of 3e-5 and no L2-regularization. The models were trained for 12 epochs with a batch size of 16 with the Adam optimizer. In addition to the accuracy, we report the macro F1-score to evaluate the performance on rare species (see Koh et al. (2021a) for details). All results were aggregated over three runs.

**RxRx1**   In biomedical research areas such as genomics or drug discovery, high-throughput screening techniques generate a vast amount of data in several batches. Because experimental designs cannot fully mitigate the effects of confounding variables like temperature, humidity, and measurements across batches, this creates heterogeneity in the observed datasets (commonly known as batch effect). The RxRx1 dataset comprises images obtained by fluorescent microscopy from 51 domains (disjoint experiments): training (40,612 images, 33 domains), validation (9,854 images, 4 domains), and test (34,432 images, 14 domains). The aim is to classify one of 1,139 genetic treatments. All results were aggregated over three runs.

We conducted the RxRx1 experiments in accordance with the specifications made in (Koh et al., 2021a). As for the FE, we, thus, used the ResNet50 pre-trained on ImageNet (He et al., 2016). We trained the models using AdamW with default parameters $\beta_1 = 0.9$ and $\beta_2 = 0.999$ using a learning rate of 1e-4 and a L2-regularization with strength 1e-5 for 90 epochs with a batch size of 75. We scheduled the learning rate to linearly increase in the first ten epochs and then decreased it following a cosine rate. For training the Gated Domain layer, we chose the same parameters as for the ERM. All results were aggregated over three runs.

**OBG-MolPCBA**   In biomedical research, machine learning has the potential to accelerate drug discovery while reducing the experimental overhead due to lowering the number of experiments required. However, to leverage the potential of machine learning, the models need to generalize to molecules structurally different from those seen during training. To study this OOD generalization across molecule scaffolds, we use the OGB-MolPCBA dataset. This dataset is split into the following subsets according to the scaffold structure: training (44,930 domains), validation (31,361 domains), and test (43,739 domains). The task is to classify the presence/absence of 128 biological activities based on a graph representation of a molecule.

In line with Koh et al. (2021a), we use a Graph Isomorphism Network (GIN) combined with virtual nodes (GIN-virtual; Xu et al. (2019) and Gilmer et al. (2017)) as the FE. For training ERM and our DG, we use the default parameters: five GNN layers with a dimensionality of 300 and a learning rate of 0.001. We train for 100 epochs using early stopping. As for the performance, we report the mean and standard deviation of the average precision across all scaffolds (domains) over three runs.

**CivilComments**   In the last decades, users have generated a vast amount of text on the Internet, some of which contain toxic comments. Machine learning has been leveraged for automatic text review to flag toxic comments. However, the models are prone to learn spurious correlations between toxicity and information on demographics in the comment, which causes the model performance to drop in specific subpopulations. To study this OOD task, we leverage the modified CivilComment dataset from Koh et al. (2021a). Based on text input, the task is to predict a binary label, toxic or non-toxic. The domains are defined according to eight demographic identities: male, female, LGBTQ, Christian, Muslim, other religions, Black, and White. All comments were randomly split into a disjoint training (269,038 comments), validation (45,180 comments), and test (133,782 comments) set.

Again, we follow Koh et al. (2021a) and use a DistillBERT-base-uncased model (Sanh et al., 2020) with the following parameters: batch size = 16, learning rate = 1e-5, AdamW optimizer, number of epochs = 5, L2 regularization 0.01, and the maximum number of tokens of 300. We use these default parameters for training our Gated Domain layer. The performance is measured in the worst-group accuracy and we report mean and standard deviation across five runs.

**PovertyMap**   As the FMoW example shows, satellite images in combination with machine learning models can been used to monitor sustainability and economic challenges on a global scale. Another application of these satellite images is poverty estimation across different spatial regions. However, there exists a lack of labels for developing countries since obtaining the ground truth is expensive, which makes this application attractive for machine learning models. To study the OOD generalization to unseen countries, we use a modified version of the poverty mapping dataset of WILDS (Koh et al., 2021a). The task is to predict a real-valued aset wealth index between 1 and 5 based on a multi-spectral satellite image. The domain refers to the country and whether the the the image is from a rural or urban are. In contrast to the other datasets, this dataset is split in five different folds, whereby in each fold the the training, valdiation and test set contains a disjoint set of countries, however, data from both rural and urban regions. The avergae size of each set across the 5 folds is for the training ~10,000 images (13-14 countries), ~4,000 images (4-5 different countries), and for the test set ~4,000 images (13-14 countries).

We follow Koh et al. (2021a) and use a pre-trained ResNet-18 model minimizing the sqarred error loss. For the optimization, we rely on the Adam optimizer with the following parameters: learning rate of 1e-3 with a decay of 0.96 per epoch, batch size of 64 and early stopping based on the OOD evaluation score. For evaluation, we report the Pearson correlation (r) between the predicted and actual asset wealth indices across the five different folds.

### B.2.2   On the challenge of obtaining domain labels.

In the example of hospitals (e.g. Camelyon17 dataset), domain labels come, in fact, for free. However, other examples, such as the CivilComments dataset, show the opposite. This dataset requires additional annotations (i.e., demographic identities), which can be tedious to obtain in practice. Some algorithms need these domain annotations to achieve superior performance on each subgroup. Furthermore, the task of subgroup detection in itself is a difficult and relevant problem. Coming back to our hospital example, even people from the same hospital might belong to different subpopulation (e.g. gender, race, age) and these demographic subgroups are often more relevant for diagnosis than which hospital a patient comes from. This information, however, is not always available (due to anonymization standards, for instance) and, therefore, the relevant domain annotation might be hard to obtain.

### B.2.3   General benchmark methods

Following the WILDS benchmarking procedure (Koh et al., 2021a), we compare our proposed Gated Domain layer to the following baselines. First, empirical risk minimization (ERM), which minimizes the average training loss over the pooled dataset. Second, a group of DG algorithms provided by the WILDS benchmark, namely, Coral, Fish, IRM, and DRO.

The Coral algorithm introduces a penalty for differences in means and covariances of the domains feature distributions (i.e., the activation of the last layer of neural networks) between the source and test domain. Thus, Coral is representative of the class of methods that promote the learning of feature *representations that*

*have similar distributions across domains.* The Fish algorithm achieves DG by approximating an inter-domain gradient matching objective, i.e., maximizing the inner product between gradients from different domains (Shi et al., 2021). Conceptually, Fish learns feature *representations that are invariant across domains.* Invariant risk minimization (IRM) introduces a penalty for feature distributions with different optimal classifiers for each domain (Arjovsky et al., 2019). The idea is to enable generalization by learning domain-invariant causal predictors. Lastly, group distributionally robust optimization (DRO) explicitly minimizes the training loss on the worst-case domain (Sagawa et al., 2020; Hu et al., 2018). This training method is an example of distributionally robust optimization (DRO). It aims to optimize for the potential test distributions' worst-case loss.

In addition to the baselines originally presented in Koh et al. (2021a), we consider the following more recent DG baselines. First, we describe LISA, which instead of regularizing the internal representations for generalization, seeks to learn domain-invariant predictors with selective data augmentation Yao et al. (2022). Conceptually, Common Gradient Descent (CGD), introduced by Piratla et al. (2022), emphasizes groups that enable better performance, thus differs from Group DRO that focuses on groups with the highest regularization. CGD, technically speaking, focuses on shared gradients that result in improvement for all groups throughout the training process. Last, Adaptive Risk Minimization using batch normalization (ARM-BN) by Zhang et al. (2021) is different from the methods presented since it adapts to previously unseen domains during test time using unlabeled observations from this test domain.

