# OpenReview forum: "Gated Domain Units for Multi-source Domain Generalization"
_TMLR — Accepted by TMLR_

### Review · Reviewer_4JDc · 2023-06-13

**Summary Of Contributions:**

This paper presents a new domain generalization (DG) method build on the idea of invariant elementary distributions. In short, the latter postulates that source and test domains are all composed of elementary distributions that remain invariant across them. This is contrary to other methods that seeks an invariant representation space that is invariant over the whole basis and not only over some of its elements. The authors further propose a gated unit to decompose test time domain into a convex combination of such IEDs using a similarity between them in the kernel mean embedding space. The results presented in the paper show that the method works and can provide a good OOD generalization on challenging datasets.

**Audience:**

Yes

**Claims And Evidence:**

No

**Requested Changes:**

**Major changes**

1. I would really encourage the authors to revise their presentation of the theoretical results. After rereading Proposition 3.1 multiple times, I still do not understand how it connects to the algorithm that is proposed. In case of Proposition 4.1 it is much clearer but for the part on Pareto optimality I struggled to understand its importance.

2. I feel that the experimental part of the paper can be made more clear with a more clear presentation of the baselines. I had to go through the text multiple times to see how the baselines differ one from another.

**Minor changes**

The manuscript requires a thorough proofreading as it is currently contains a fair amount of typos and errors.

1. The motivation is to become independence of domain labels -> independent
2. The second challenge is the lack of effective architectures for these approaches, particularly ones that can be easily integrated into existing deep learning frameworks -> I'm not sure  that this is true, seems like the authors may want to downgrade this claim a bit
3. In Tale 2,  -> Table 2
4. reveals: M -> reveals methods
5.  Davies-bouldinn Score -> Davies-Bouldin

**Strengths And Weaknesses:**

**Strengths**

1. Interesting idea that seems to be novel;
2. Simple, yet efficient implementation that can be used with different backbones;
3. Theoretical justification
4. Reasonable effectiveness

**Weaknesses**

1. Link between the theory and implementation is somewhat unclear
2. Not clear whether the method is compared to a simpler baseline decomposing each test domain as a convex combination of source domains
3. The results are good although the proposed method requires some ad-hoc routines to set its parameters

---

> ### Author Response · Authors · 2023-07-24
> **Response to Reviewer 4JDc**
>
> We would like to thank the Reviewer for the positive feedback and the constructive comments on our manuscript, which will improve the overall quality of our work.
>
> **Weaknesses**
>
> >Link between the theory and implementation is somewhat unclear
>
> We thank the Reviewer for highlighting th unclarities in our work. We adressed this weakness in the first comment of the Paragraph major changes.
>
> >Not clear whether the method is compared to a simpler baseline decomposing each test domain as a convex combination of source domains
>
> Following [1,2], we compared our methods against the ERM baseline trained on the combined source data, one of the strongest baselines in domain generalization. In addition, we compared our method against an ensemble of such ERM models, where the ensemble components are jointly trained on the combined source data. The number of ensembles in this experiment is equal to the number of source domains. Since the ERM model trained on the combined source data remains one of the strongest baselines, we consider an ensemble of ERM models to be a simpler but more effective baseline than an ensemble of models trained on each domain individually. Nevertheless, we are happy to include other baselines at the Reviewer's request.
>
> >The results are good although the proposed method requires some ad-hoc routines to set its parameters
>
> We agree with the Reviewer and do not want to diminish our ad hoc parameter-setting routines. Related work also requires setting their hyperparameters [as pointed out in the review/benchmarking 1,2], for example, based on grid search. However, our heuristics for the main parameters are consistent with our theoretical motivation. Of note: Our method is also compatible with conventional hyperparameter optimization tools, making the selection more conventional for potential applications in research and practice. Nevertheless, in our work, we would not have fully understood whether the performance improvements were due to our methods or extensive hyperparameter optimization. Therefore, we relied on reasonable heuristics instead of extensive hyperparameter tuning.
>
> **Major changes**
>
> >I would really encourage the authors to revise their presentation of the theoretical results. After rereading Proposition 3.1 multiple times, I still do not understand how it connects to the algorithm that is proposed. In case of Proposition 4.1 it is much clearer but for the part on Pareto optimality I struggled to understand its importance.
>
> We thank the Reviewer for pointing us toward this unclarity in our manuscript. We revised this part and added a thorough discussion regarding Proposition 3.1 and its connection to our method's design.
>
> In short, this Proposition is aimed to motivate the proposed I.E.D assumption and the benefits of considering such a decomposition. For example, the set of Pareto-optimal solutions is a strict subset of the hypothesis space. Therefore, searching for the optimal function within the Pareto set narrows the search space. Also, Proposition 3.1 shows how the original minimization problem can be decomposed into the optimization of $K$ risk functionals $(R_1, \ldots, R_K)$ which motivates a modular structure of our layer and introducing Gated Domain Units that optimize these functionals. Finally, the Pareto optimality set is independent of the actual values of coefficients $\alpha$, which makes it indeed invariant w.r.t. $\alpha$ and allows us to combine a solution for any target distribution from the members of the Pareto set. This result motivates the ensemble-like form of the final solution where the $\alpha$ coefficients are derived based on the similarity between the target and elementary distributions.
>
> >I feel that the experimental part of the paper can be made more clear with a more clear presentation of the baselines. I had to go through the text multiple times to see how the baselines differ one from another.
>
> We thank the Reviewer for suggestions on how to improve our presentation of the basic methods in Section 5 (Experiments). We have revised this section and the corresponding Appendix. Specifically, we have added a clear description of the baselines for the Digits experiment (ERM and an ensemble of ERM models, where the number of components $M$ equals the number of GDUs) in Section 5.1.2.
>
> The experiments in Section 5.2 include nine baselines and eight data sets. Due to the overall page limit, we have described the baseline methods, parameters, and datasets in the Appendix (Sections B.2.1 and B.2.3) to ensure reproducibility, as referenced in the main text. To improve our presentation, we have added a short overarching summary of the baseline methods to the main text. However, we suggest leaving the main descriptions in the Appendix to keep the main text short.

---

> > ### Author Response · Authors · 2023-07-24
> > **Response to Reviewer 4JDc - References**
> >
> > **References**
> >
> > [1] Koh, P. W. et al. WILDS: A Benchmark of in-the-Wild Distribution Shifts. in Proceedings of the 38th International Conference on Machine Learning (eds. Meila, M. & Zhang, T.) vol. 139 5637--5664 (PMLR, 2021).
> >
> > [2] Gulrajani, I. & Lopez-Paz, D.In Search of Lost Domain Generalization. 29 (2021).

---

### Review · Reviewer_2Zu3 · 2023-07-09

**Summary Of Contributions:**

The submission studies the problem of domain generalisation, a particular instantiation of the out-of-distribution generalisation class of problems. A new modelling assumption about the structure of novel distributions, and how they relate to distributions seen during training, is proposed. It is a strict generalisation of some previous work in this area. A new neural network layer is developed that leverages this modelling assumption. Experiments are conducted on a number of DG benchmark datasets, measuring both performance on unseen domains and as investigations into the robustness of the method to choices of hyperparameters and inclusion of different components in the method.

**Audience:**

Yes

**Broader Impact Concerns:**

It is hard to predict direct broader impact from this work, as it is quite foundational. Some aspects of potential broader impact are discussed in the introduction section while explaining why DG is a useful problem to study.

**Claims And Evidence:**

Yes

**Requested Changes:**

Additional analysis of whether the modelling assumption holds in practice is welcome, but not required for acceptance.

I would recommend tweaking the first sentence of Section 2. Out-of-distribution generalisation is a much broader term than encompasses specific problem settings such as domain adaptation and domain generalisation, among many others.

**Strengths And Weaknesses:**

The paper is very clearly written and provides ample relevant links to previous literature in the area. I found the paper very easy to read.

The proposed modelling assumption is novel, as far as I am aware. Given that it is a strict generalisation of previous work that has assumed novel domains are a convex combination of seen domains, this new assumption makes sense. I.e., treating each domain as a mixture distribution and having novel domains be a novel combination of the underlying mixture components seems like a sensible approximation of the underlying problem.

The proposed neural network component is simple and well-motivated, having clear connections to previous work in kernel machines. The ability to train it with standard learning algorithms is also a bonus.

The experimental evaluation is quite thorough, and the analysis is level-headed and informative. The ablations, such as the analysis of the heuristic for selecting the number of latent domains, are useful.

Something that would be nice to include, but I am not sure how feasible it is, would be an investigation into the extent to which the proposed modelling assumption holds in practice. An easy way to partially address this might be to demonstrate that there are plausible scenarios where it is accurate, via constructing synthetic data for a simple but realistic problem. Ideally it would be good to demonstrate that the assumption holds on much more realistic data, but I am not sure this is possible.

---

> ### Author Response · Authors · 2023-07-24
> **Response to Reviewer 2Zu3**
>
> We thank the Reviewer for the positive feedback and constructive comments on our modelling assumptions, which will enhance our work's overall quality.
>
> >Additional analysis of whether the modelling assumption holds in practice is welcome, but not required for acceptance.
>
> We would like to point to Figure 1, which illustrates the assumption of elementary domains in the real world. We used a k-means clustering approach on the camelyon17 dataset, where the objective is to generalize well on an unseen hospital. We observe that the hospitals share the same clusters with similar visual characteristics (e.g., illumination). We considered this example as a real-world illustration of our I.E.D assumption. In addition, we analyzed what the elementary domains represent in the joint multi-source dataset in Figure 9.
>
> Nonetheless, we agree that additional experiments showing if and under which conditions the I.E.D assumption holds in practice would be valuable. Therefore, we are currently working on whether we can run additional experiments as instructed by the Reviewer. However, we can not guarantee that these experiments will be done by the end of the rebuttal period.
>
> >I would recommend tweaking the first sentence of Section 2. Out-of-distribution generalisation is a much broader term than encompasses specific problem settings such as domain adaptation and domain generalisation, among many others.
>
> Thank you for pointing out the unclarity. We revised the manuscript accordingly and do not use out-of-distribution as a synonym for domain generalization.

---

### Review · Reviewer_o5Po · 2023-07-11

**Summary Of Contributions:**

This paper considers the domain generalization issue under the Invariant Elementary Distribution (I.E.D) assumption.

From theory, the paper argues why the IED assumption is important and some novel theoretical results based on IED assumption.

From algorithm, the paper designs GDUs to present the effectiveness of their idea.



**Audience:**

Yes

**Claims And Evidence:**

Yes

**Requested Changes:**

Given more details on Definition 1.

More experiments to support the proposed methods.

More references to support the paper.

**Strengths And Weaknesses:**

Strengths:
1. This is a novel idea.
    I like the idea. In fact, I want to use the same idea (IED assumption) to do my work last year.

2. Clear presentation and organization.

Weakness:
1. I am little confused on Definition 1. There are no such f st R(g)>R(f). That is means R(f) >= R(g)? Is there some issues here?

2. The exmperiments are week. Maybe more experiements are required.

3. How to select the basis Vj?

4.  Missing references:
    Please cite
    1) Moderately Distributional Exploration for Domain Generalization, ICML 2023
    2) Learning Causally Invariant Representations for Out-of-Distribution Generalization on Graphs

In summary: I think the idea of this paper should be carried forward. I like it from the theoretic views.

---

> ### Author Response · Authors · 2023-07-24
> **Response to Reviewer o5Po**
>
> We thank the Reviewer for the comments and helping us to improve the overall quality of our work.
>
> >I am little confused on Definition 1. There are no such f st R(g)>R(f). That is means R(f) >= R(g)? Is there some issues here?
>
> That would have been true in a one-dimensional case. In a case of $K \geq 2$, however, not every two hypothesis $f, g \in \mathcal{F}$ can be ordered, e.g. when $R_1(g) > R_1(f)$ and $R_2(g) < R_2(f)$. Therefore, $f \in \mathcal{F}$ is Pareto optimal if there exists no hypothesis $g \in \mathcal{F}$ that optimizes *every* risk functional $R_j$ better than $f$. We have included this comment in the manuscript.
>
> >The exmperiments are week. Maybe more experiements are required.
>
> We would be very grateful to hear more details about which experiments are weak and how we could improve them.
>
> >How to select the basis Vj?
>
> The basis vectors $V_j$ are not chosen but learned from the data. They are stored as a weight matrix of each GDU and trained via backpropogation jointly with the learning machine $f(\tilde{x}, \theta_j)$ as mentioned in Section 4.
>
> >Missing references: Please cite
> *   Moderately Distributional Exploration for Domain Generalization, ICML 2023
> *   Learning Causally Invariant Representations for Out-of-Distribution Generalization on Graphs
>
> >In summary: I think the idea of this paper should be carried forward. I like it from the theoretic views.
>
> Thank you very much for your positive opinion about our work. We have added the missing refernces to our manuscrtip.
>
> **Requested Changes:**
> >Given more details on Definition 1.
>
> We have revised the Definition 1 and added the comment from below (response to Revier 4JDc) to clarify the meaning of Definition 1.
>
> >More experiments to support the proposed methods.
>
> We thank the Reviewer for this comment. We would appreciate further suggestions on which experiments could strengthen our claims.
>
> >More references to support the paper.
>
> In Section 2, we reviewed related work closely related to our method. Specifically, we focused on prior work that builds on the concept of latent domains to highlight how our work differs from and thus complements this line of research. However, we agree that this review can be expanded. To do so, we have now reviewed work from another but related line of research called ensemble learning. We have added and discussed the references [3,4,5] in Section 2.
>
> Additionally, we have also included the above references in our manuscript. We hope that the discussion of these additional references is considered sufficient. Otherwise, we are happy to include additional suggested references.
>
> [3] Y. Zhang, J. Wang, J. Liang, Z. Zhang, B. Yu, L. Wang, X. Xie, and D. Tao, “Domain-specific risk minimization for out- of-distribution generalization,” arXiv preprint arXiv:2208.08661, 2023.
>
> [4] Y. Yao and G. Doretto, “Boosting for transfer learning with multiple sources,” in Proc. IEEE Comput. Soc. Conf. Comput. Vis. Pattern Recognit., San Francisco, CA, USA, Jun. 2010, pp. 1855–1862.
>
> [5] J. Gao, W. Fan, J. Jiang, and J. Han, “Knowledge transfer via multiple model local structure mapping,” in Proc. 14th ACM SIGKDD Int. Conf. Knowl. Discovery Data Mining,LasVegas,NV,USA, Aug. 2008, pp. 283–291.

---

### Author Response · Authors · 2023-07-24
**Revised Manuscript Uploaded by 25th July, 12 PM AoE.**

We thank all Reviewers for investing a significant amount of time to review our work and provide positive comments that help improve our work's overall quality.

We will upload the revised manuscript latest by tomorrow, 25th July, 12 PM AoE.

---

### Author Response · Authors · 2023-07-25
**Revised Manuscript Now Uploaded**

We have submitted the revised version of our manuscript and made sure to address any remaining comments. All major changes have been highlighted in blue and will be formatted accordingly upon acceptance.

We hope that the reviewers and editors will find our work ready for publication. If there are any remaining comments, we are willing to address them for the camera-ready version.

---

### Decision · Action_Editors · 2023-08-28

**Recommendation:** Accept with minor revision

**Comment:**

This paper addresses the problem of domain generalization. Based on the Invariant Elementary Distribution (IED) assumption, which represents new domains as a convex combination of seen domains, the authors introduce Gated Domain Units (GDUs) to learn the representation for each latent elementary distribution. Experiments are conducted on Digits datasets and on standard datasets for domain generalization, i.e., DomainBed and WILDS. The paper initially received positive feedback. The reviewers appreciated the relevance of the proposal, the theoretical analysis, and the experiments. However, the reviewers also raised concerns about the validity domain of the IED assumption, insights into the theoretical results, and clarification on experiments for the baselines or hyper-parameter tuning. The authors' feedback, discussions, and the revised manuscript convinced the reviewers, who unanimously recommended the paper for acceptance.

The AE has carefully read the submission and the discussions. The AC considers that the approach is sound and relevant, and that the IED model and the proposed method to learn it are validated by evidence. The AE also appreciates the quality of the paper's presentation and the inclusion of the changes that have been requested by the reviewers. Therefore, the AE recommends acceptance. As a minor suggestion, the AE recommends that the authors clarify whether they will include new results to assess the validation of the IED assumption, as requested by R2Zu3 (though it is not required for acceptance).

**Audience:**

The submission deals with domain generalization, a major problem in machine learning that should be of interest for a wide TMLR audience.

**Claims And Evidence:**

The claims for the proposed methods on domain generalization and the related assumptions are clearly stated. The claims are validated by evidence with a convincing set of experiments.

---

> ### Author Response · Authors · 2023-09-22
> **Camera-ready version uploaded**
>
> We thank the Editorial Team and Reviewers for investing the time to review our work and helping us with constructive comments to improve our work further. We are thankful for the chance to publish our work at TMLR. We have now uploaded the camera-ready version.
>
> We would like to give our final comment on the additional simulation experiment: We discussed simulating multi-domain data using Gaussian mixtures for the simulation experiments, and investigating whether the GDUs can approximate the components. However, this experiment is quite similar to the digits data experiments, and the contribution is not significant enough to justify a separate section in the paper, including another round of discussion. Hence, we decided to publish the paper without additional simulation experiments.